# CANDERE-COACH: Reinforcement Learning from Noisy Feedback

**Yuxuan Li[1,2]**[*]                                                    *yuxuan.li1@uwaterloo.ca*
**Srijita Das[3]**[†]                                                         *sridas@umich.edu*
**Matt Taylor[2,4]**[†]                                          *matthew.e.taylor@ualberta.ca*

[1] *David R. Cheriton School of Computer Science, University of Waterloo*
[2] *Department of Computing Science, University of Alberta*
[3] *Computer and Information Science Department, University of Michigan–Dearborn*
[4] *Alberta Machine Intelligence Institute*

**Reviewed on OpenReview:** *https://openreview.net/forum?id=JgP7Wepetn*

## Abstract

Reinforcement learning (RL) has been widely applied to many challenging tasks. However, in order to perform well, it requires access to a good reward function, which is often sparse or manually engineered with scope for error. Introducing human prior knowledge is often seen as a possible solution to the above-mentioned problem, such as imitation learning, learning from preference, and inverse reinforcement learning. Learning from feedback is another framework that enables an RL agent to learn from binary evaluative signals describing the teacher's (positive or negative) evaluation of the agent's action. However, these methods often make the assumption that evaluative teacher feedback is perfect, which is a restrictive assumption. In practice, such feedback can be noisy due to limited teacher expertise or other exacerbating factors like cognitive load, availability, distraction, etc. In this work, we propose the CANDERE-COACH algorithm, which is capable of learning from noisy feedback by a sub-optimal teacher. We propose a noise-filtering mechanism to de-noise online feedback data, thereby enabling the RL agent to successfully learn with up to 40% of the teacher feedback being incorrect. Experiments on three common domains demonstrate the effectiveness of the proposed approach.

## 1 Introduction

Reinforcement learning (RL) has made rapid progress in part due to the advancements of deep learning, which has been successfully applied to the challenging game of Go (Silver et al., 2017), solving Rubik's cube with a robot arm (Akkaya et al., 2019), and for deciding the treatment regimen for cancer (Tseng et al., 2017). Although successful in solving these problems, its progress has still been significantly hindered by the well-known sample-inefficiency problem (Ibarz et al., 2021) because the agent may need millions of interactions with an environment to learn a near-optimal policy.

Moreover, deep RL's performance often deteriorates in domains with sparse reward because of slower back-propagation of the reward signal to the entire state-space (Yu et al., 2020; Knox et al., 2023). To combat this problem, reward functions may be manually specified by task experts or RL developers, often by trial-and-error. However, even when human subjects carefully designed reward functions, errors can lead to reward hacking (Laidlaw et al., 2024) or other unintended behaviors such as reward overfitting (Booth et al., 2023). To address sample inefficiency and problems related to reward design, human-in-the-loop RL (Retzlaff et al., 2024) has been used to guide RL algorithms. Human prior knowledge has been used in the form of

---

[*]Corresponding author.
[†]Equal advising.

demonstrations (Schaal, 1999), action-advice (Torrey & Taylor, 2013), policy-shaping (Griffith et al., 2013), to name a few. To address the reward design problem in RL, human advice has been used to learn the reward model in various frameworks, such as inverse reinforcement learning (Ng et al., 2000), learning from preference (Christiano et al., 2017; Lee et al., 2021), learning from human feedback (Knox & Stone, 2009; MacGlashan et al., 2017), etc. Deep COACH (Arumugam et al., 2019) is one such learning from feedback paradigm in which the teacher observes an agent in action and provides scalar feedback denoting agreement or disagreement. This feedback is, in turn, used as an advantage function in the RL algorithm. Although this method has been successful in training RL agents without a reward function, one major assumption is that the feedback is perfect (e.g., noise-free). However, collecting feedback from humans is an expensive process that demands attention, time, and focus from the human. Hence, this assumption is restrictive, and the collected feedback might be noisy. Addressing this critical shortcoming and making feedback learning more useful in real-world settings is the primary motivation of this work.

Although there has been work in the RL literature to handle imperfect human knowledge (particularly with respect to demonstrations (Chen et al., 2021)) and preferences Cheng et al. (2024), learning from noisy binary feedback is an important advice modality that has not been addressed in the RL literature. In contrast, the identification of noisy data (Han et al., 2018; Younesian et al., 2021) and outlier detection (Liu et al., 2013) has been widely studied in supervised learning. Motivated by these advances, we propose a learning from feedback framework where 1) the agent does not have access to the reward function, 2) can learn from *noisy feedback* given by the human/agent teacher as positive and negative feedback, and 3) the agent learns a policy to maximize the likelihood of the teacher's objective. Our proposed framework consists of a *classifier augmented noise detecting module* that is capable of detecting and filtering noisy feedback, which is further used to guide the RL agent to solve the task.

**Contributions** of this work include: (1) demonstrating that well-known learning from feedback methods like Deep COACH can fail to learn from noisy, limited binary teacher feedback; (2) introducing a noise detection model to identify and filter noise within this framework, (3) investigating the behavior of the proposed algorithm with varying feedback noise levels; (4) empirically evaluating three domains, showing that our proposed method learns from teacher feedback with up to 40% noise, significantly outperforming the relevant baselines. We also show the potential for using this method as a plug-and-play tool with other learning from binary feedback methods like Deep TAMER. The codebase can be found in `https://github.com/liyuxuan-academic/candere-coach`.

## 2 Related Work

**RL agents learning from teacher advice:** RL agents are often challenged by sparse reward domains. Receiving help from a knowledgeable teacher can ameliorate such problems, such as when a teacher agent (or human) provides demonstrations (e.g., in imitation learning (Billard et al., 2003; Giusti et al., 2015; Ross et al., 2011) or inverse reinforcement learning (Das et al., 2021; Ho & Ermon, 2016)). There are also teacher-student frameworks (Torrey & Taylor, 2013; Ilhan et al., 2021) where the student agent asks for action advice from the teacher, and learning from preferences (Wilson et al., 2012; Christiano et al., 2017; Lee et al., 2021), where preferences over pairs of trajectories are provided. Learning from feedback is another advising framework, where agents learn from binary feedback provided by the teacher. Knox et al. (Knox & Stone, 2009) proposed TAMER to exploit human feedback signals, which in turn is used to learn an expectation of human feedback that the agent can maximize. Macglashan et al. (MacGlashan et al., 2017) proposed COACH, which uses feedback as the advantage function in the policy-gradient objective. These methods have also been extended in Deep RL settings like Deep TAMER (Warnell et al., 2018) and Deep COACH (Arumugam et al., 2019). Loftin et al. (Loftin et al., 2016) proposed methods that take the teacher's strategy into account by inferring the teacher's strategy, but most of the experiments are done in bandit domains. Compared to learning from demonstration methods, learning from feedback may require less knowledge, as judging good or bad actions is essentially easier than specifying the optimal action. However, to the best of our knowledge, all previous work on learning from feedback does not explicitly try to detect and correct errors. We propose a learning from feedback method that learns from *noisy teacher feedback*.

**Noise detection in supervised learning:** Learning with noisy data has become increasingly challenging with massive datasets and increasing the risk of noise from annotated data collection (Song et al., 2022). In supervised learning, noise can be easily memorized by the deep networks (Zhang et al., 2021), hurting generalization performance. Several learning methods have been proposed to detect noisy labels. Mentor-Net (Jiang et al., 2018) uses a separate mentor network to guide the classifier with a curriculum of noisy labels. For semi-supervised anomaly detection, (Chen et al., 2015) uses SVDD as a one-class classifier to detect outliers. (Han et al., 2018) proposed Co-teaching, an example of an unsupervised anomaly detection method. Co-teaching maintains two deep networks and lets them select data based on their respective losses and feed outputs into each other reciprocally, allowing learning from noisy data. Similar ideas of selecting data based on losses can be found in QActor (Younesian et al., 2021), where detected noisy labels are further corrected by an oracle during active learning. Motivated by progress in this direction, our proposed work adapts loss-based noise detection techniques for deep networks into a learning from feedback framework for training deep RL agents from noisy feedback.

**Noise-robust RL from teacher advice:** Recently, RL learning from preferences has become a popular paradigm that does not require access to a reward function. Instead, a reward function is learned from comparative preference over pairs of trajectories (Christiano et al., 2017). Human teachers inevitably provide noisy labels, and this form of learning paradigm suffers when noisy preferences are provided. (Cheng et al., 2024) proposed RIME, which detects noisy preference by measuring entropy in predictions of inferred reward functions. (Xie et al., 2025) proposed the HSBC framework that detects noise by hypothesis space cutting, which avoids inconsistent preferences. Furthermore, there has been an increasing trend of learning from language model preference (Wang et al., 2024). In RL-SaLLM-F (Tu et al., 2024), a double-check mechanism depending on LLM's self-consistency is proposed to filter noisy VLM preference . (Huang et al., 2025) proposed a tri-teaching framework to detect noisy human or VLM preference, in addition to an additional imitation learning loss to further strengthen the robustness of learning. However, all of the above-mentioned work focuses on detecting noise in learning-from-preference settings, where feedback is provided over pairs of trajectories. In contrast, our work investigates the problem of noisy feedback in established human-in-the-loop frameworks such as COACH and proposes an algorithm to mitigate the impact of noise in these settings. As a result, we focus on discrete action domains where binary feedback is preferred compared to continuous control tasks, where all the above-mentioned works are applicable. Reinforcement Learning with Human Feedback (Ouyang et al., 2022; Dai et al., 2023) has recently emerged as a popular paradigm for aligning pretrained language models. Although RLHF is closely related to our work in its use of feedback for learning, the underlying learning dynamics differ substantially when applied to pretrained large language models in comparison to small models from scratch. We therefore consider RLHF outside the scope of this paper.

## 3 Background

**Reinforcement Learning:** RL is defined using a Markov Decision Process (MDP). An MDP is denoted by a quintuple as $M = \{\mathcal{S}, \mathcal{A}, \mathcal{T}, r, \gamma\}$, where $\mathcal{S}$ denotes the agent's state space, $\mathcal{A}$ is the agent's action space, $\mathcal{T} : \mathcal{S} \times \mathcal{A} \times \mathcal{S} \to [0, 1]$ is the environmental dynamics transition probability, $r : \mathcal{S} \times \mathcal{A} \times \mathcal{S} \to \mathbb{R}$ is the function that gives an immediate reward, $\gamma$ is a discount factor. In a typical MDP setting, the RL agent starts at state $s_0$ and takes an action $a_0$ following its policy $\pi$, which leads the agent to its next state $s_1$ and receives reward $r_0$. The interaction repeats for $T$ steps until a terminal state is reached. RL agent optimizes the policy $\pi$ by maximizing the expected cumulative reward $\mathbf{E}_{\tau \sim \pi}[\Sigma_{t=0}^{T} r_t]$

**Deep COACH:** Feedback is often defined as a scalar value $f$, describing the teacher's judgment of the agent's current behavior, and the teacher can be a human teacher or a scripted teacher, like COACH (Warnell et al., 2018) and TAMER (Knox & Stone, 2009), we define feedback as $f \in \{-1, 1\}$, where $-1$ means the teacher discourages the agent's behavior, and 1 encourages the agent's behavior, which is a pair of state and action $\langle s_t, a_t \rangle$, at time step $t$. In COACH, the feedback $f_t$ is deemed as an estimate of the advantage value and is used to update the policy. In Deep COACH (Arumugam et al., 2019), the policy is updated as Equation 1:

$$\nabla_{\theta_t} J(\theta_t) = \mathbf{E}_{a \sim \pi_{\theta_t}^h(\cdot|s_t)}[\nabla_{\theta_t} log(\pi_{\theta_t}(a_t|s_t)) \cdot f_t]. \tag{1}$$

## 4 Classifier Augmented Noise Detecting and Relabelling COACH

### 4.1 Problem Statement

**Given:** An RL agent that has no access to the reward function $r$, a teacher $T$ that can provide noisy binary feedback $\hat{f} \in \{-1, 1\}$ based on the learning agent's visited state and action; and the ground truth feedback $f \in \{-1, 1\}$.

**Objective:** Train the agent policy $\pi_\theta$ by using the noisy feedback $\hat{f}$ given by the teacher.

**Assumptions:** We make the following assumptions: (1) Static and symmetric noise is added to $f$ to produce $\hat{f}$ (i.e., the noise distribution does not change over time and every feedback can be incorrectly flipped with equal probability). (2) The proportion of labels that are flipped (i.e., the noise proportion $p_{noise}$) is known.* (3) The ground truth feedback is deterministic for each state and action pair, i.e., only one correct feedback exists for each state-action pair. (4) The agent does not have access to the reward function during training (but may be used to evaluate the agent's policy).

### 4.2 Methodology and Algorithm

The proposed framework **C**lassifier **A**ugmented **N**oise **DE**tecting and **RE**labelling COACH (CANDERE-COACH) is illustrated in Figure 1. The agent follows the policy $\pi_\theta$. The teacher provides noisy binary feedback $\hat{f}$ on the state-action pairs visited by the agent. This feedback, along with the corresponding state-action pair, is stored in a replay buffer. During training, a mini-batch is sampled and is input to a classifier, $C_\phi$ (Section 4.2.1), which acts as a noise detector. This classifier is trained (as the agent learns) and identifies correct and incorrect feedback based on its loss function (Section 4.2.2). We also employ a method called 'Active Relabeling' (Section 4.2.3) to convert some of the feedback suspected to be incorrect. The aggregation of the original and relabelled feedback constitutes the filtered batch — this batch becomes input for the agent's policy training and the training of the classifier. CANDERE-COACH comprises the following components:

**4.2.1 Noise-filtering classifier**: In order to identify noisy feedback, we train a classifier $C_\phi : S \times A \to [0, 1]$, † which maps the state and action pair to the predicted feedback probability distribution. The feedback dataset $\mathcal{D} = \{(s_i, a_i, \hat{f}_i)\}_{i=1}^{N}$ where $(s_i, a_i)$ refers to the $i^{th}$ state, action pair and $\hat{f}_i$ refers to the corresponding observed feedback. This classifier is pretrained on a small noise-free feedback dataset $\mathcal{D}$ using the cross entropy loss, as shown in Eqn 2, where $N$ refers to the size of the pretraining dataset, $P(c = \hat{f}_i | s_i, a_i, \phi)$ refers to the predicted probability of feedback by the classifier and $c \in \{-1, +1\}$ is the feedback class.

$$l(\phi) = -\sum_{i=1}^{N} \sum_{c \in \{-1,1\}} \mathbb{I}[\hat{f}_i = c] \log P(\hat{f}_i = c | s_i, a_i, \phi) \tag{2}$$

We also use focal loss (Lin et al., 2017) instead of cross entropy loss for pretraining datasets of imbalanced classes, as shown in Eqn 3. $\gamma_{focus}$ is the focusing parameter, and $\alpha : f \to \mathbb{R}$ is a map from a data class of input to a scalar weight, which is set based on the proportion of that class in the dataset.

$$l(\phi) = -\sum_{i=1}^{N} \sum_{c \in \{-1,1\}} \mathbb{I}[\hat{f}_i = c] \log P(\hat{f}_i = c | s_i, a_i, \phi) \alpha(c) (1 - P(\hat{f}_i = c | s_i, a_i, \phi))^{\gamma_{focus}} \tag{3}$$

The classifier $C_\phi$ is retrained after every step of noise detection and active relabeling, described in detail in the next section.

**4.2.2 Noise-detection**: In this step, we detect potentially noisy feedback by using the trained classifier $C_\phi$. In every learning iteration, once a mini-batch is sampled from the replay buffer, $C_\phi$ predicts the point-wise

---

*We relax this assumption later in Effect of noisy pretraining dataset, Section 5.2.

†The output is actually a vector of predicted probabilities for positive and negative feedback that sums to 1.

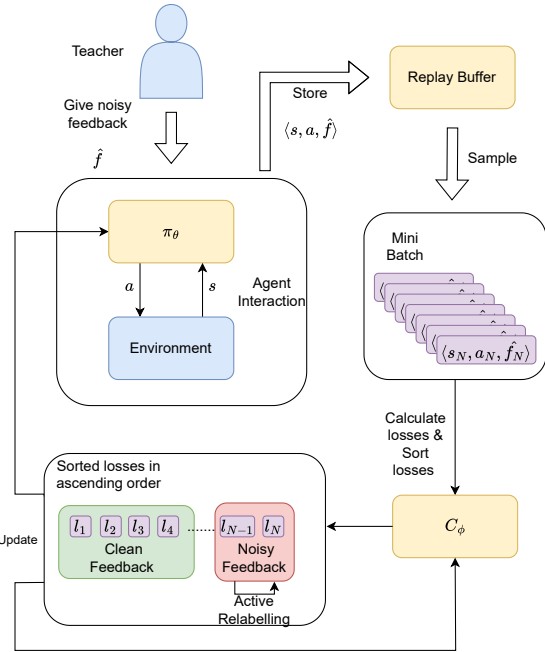

Figure 1: The overview of CANDERE-COACH . We use a classifier $C_\phi$ to filter noisy feedback and update policy $\pi_\theta$ and $C_\phi$ with filtered minibatches.

loss of each state-action-feedback tuple, which are then sorted in increasing order as per their loss values. We treat data points with small losses as clean data. This is because neural networks can learn clean data in the early stages of training Han et al. (2018) because estimating the distributions of these data-points is relatively easier. Mathematically, a clean data batch $B_c$ as identified by the classifier is filtered as mentioned below, where $l_{B'}(\phi)$ is the cross-entropy loss of the classifier, and $R(B)$ refers to the remember rate that specifies how many data points need to be included in the batch.

$$B_c = \underset{B':|B'|\geq R(B)|B|}{\arg\min} \; l_{B'}(\phi) \tag{4}$$

In the above equation, we set remember rate $R(B) = 1 - p_{noise}$. After $B_c$ is identified, we sort the data points with larger loss values and consider them as potentially noisy data because deep networks cannot estimate noisy distributions in the early stages of training, owing to these conditional distributions being harder to estimate Han et al. (2018). Hence, the suspected noisy batch $B_n$ as identified by the classifier is estimated as below:

$$B_n = \underset{B':|B'|\leq R'(B)|B|}{\arg\max} \; l_{B'}(\phi) \tag{5}$$

In the above equation, $R'(B)$ refers to a hyperparameter that decides how many data points need to be included in $B_n$. While in our implementation, $R'(B) = p_{noise}$, so that $B = B_n \cup B_c$, $R'(B)$ can be set to different values for more aggressive or conservative denoising as in prior work (Han et al., 2018).

**4.2.3 Active relabeling**: Once a suspected noisy batch $B_n$ has been identified, we flip $R_{flip}$ proportion of labels with highest losses resulting in $B_{ar}$ [‡]. The intuition is that the opposite feedback label is necessarily the correct label for binary-valued feedback. We call this method *active-relabelling*. Given a label flipping rate of $R_{flip}$, for a batch of size $|B|$, after we identified the noisy labels (i.e, $|B_n| = R'(B)|B|$), we further flip $|B_{ar}|$ labels from $B_n$, where $|B_{ar}| = R_{flip}|B_n| = R_{flip}R'(B)|B|$. After data has been relabelled, $B_c$ and $B_{ar}$ are provided to the RL agent for training the policy $\pi_\theta$ as well as to the classifier $C_\phi$ for online training

---

[‡]In our experiments, $B$ is $B_n \cup B_c$; $B_{ar} \in B_n$.

as per Eqn 2. We use policy gradient to train the RL agent using the filtered feedback as per Eqn 1.

**Algorithm:** Algorithm 1 shows the algorithm for CANDERE-COACH. With a pretrained classifier $C_\phi$, the agent interacts with the environment and queries the teacher for feedback at a set frequency until the budget $f_b$ is depleted. (line 5). The agent's observation, action, and feedback are stored as a tuple in the replay buffer $R$. After sampling a minibatch $B$ from R, it is fed into the classifier to evaluate data point-wise cross entropy loss (line 8). These loss values are sorted in ascending order, and the first $R(B)|B|$ data-points in the batch are selected as $B_c$ (line 10). Similarly, $R'(B)|B|$ data-points (denoted as $B_n$) are selected by sorting them per their cross entropy loss value (Equation 2) in descending order and the top $R_{flip}$ proportion feedback label of this set is flipped (lines 11–12) as $B_{ar}$. In the last step, lines 14–15, in addition to updating the policy $\pi_\theta$, the classifier is also updated using online training with the filtered feedback batch ($B_c$ and $B_{ar}$). The classifier gradually adapts to the new state distribution as in the training set, and related ablation study can be found in Section 5.2. In a budgeted setting, when the budget $f_b$ is depleted, the agent does not further interact with the environment as no feedback signals can be further gathered. Post the budget completion, this is an offline-RL like setting where data is sampled from replay buffer for updating the noise-aware classifier. The feedback labels of the agent transitions in the replay buffer can change based on the loss ordering of the classifier from active relabelling.

---

**Algorithm 1** **C**lassifier **A**ugmented **N**oise **DE**tecting and **RE**labelling COACH

---

**Input**: Pretrained Classifier $C_\phi$, Policy $\pi_\theta$, Teacher $T$, Noise Proportion $p_{noise}$, Maximum Episode Length $l$, Maximum # of episode $N_e$, Replay Buffer $R$ size $N$, Batch size $b$, Noise Detection Rate $R'(B)$, Remember Rate $R(B)$, Feedback Budget $f_b$, Label Flip Rate $R_{flip}$

1: Initialise random policy $\pi_\theta$
2: Initialise empty replay buffer $R$
3: **for** $i \leftarrow 1, 2, ..., N_e$ **do**
4:     **for** $j \leftarrow 1, 2, ..., l$ **do**
5:         Agent with policy $\pi_\theta$ interacts with the environment and queries teacher $T$ for noisy feedback $\hat{f}$ if budget $f_b$ not depleted
6:         Sample batch $B = \{\langle s_0, a_0, f_0 \rangle, ... \langle s_{b-1}, a_{b-1}, f_{b-1} \rangle\}$ from $R$
7:         Use Classifier $C_\phi$ to predict $f$ on state-action pairs in $B$
8:         Calculate the cross entropy loss $L = \{l_0, ... l_{b-1}\}$ for each data sample in $B$ following Equation 2
9:         Sort $B$ in ascending order based on $L$
10:        Pick $R(B)|B|$ items with minimised losses from $B$ as $B_c$, following Equation 4
11:        Pick $R'(B)|B|$ items with maximised losses as $B_n$, following Equation 5
12:        Flip feedback labels of $B_n$ with Label Flip Rate $R_{flip}$ to get $B_{ar}$
13:        Train $\pi_\theta$ with $B_c$ and $B_{ar}$ following Equation 1
14:        Train Classifier $C_\phi$ with $B_c$ and $B_{ar}$ following Equation 2
15:     **end for**
16: **end for**
17: return $\pi_\theta$

---

## 5 Experimental evaluation and results

This section addresses the following three questions:
**RQ1.** In what kind of setup is the performance of Deep COACH sensitive to noisy feedback?
**RQ2.** Can CANDERE-COACH learn effectively with different proportions of noisy feedback?
**RQ3.** Does active relabelling help CANDERE-COACH in noise correction?

**Domains:** We conduct our experiments in three Gymnasium (Towers et al., 2023) domains: Cart Pole, Lunar Lander, and Minigrid Door Key, as shown in Figure 2. In Cart Pole, the agent controls a moving cart to prevent the attached rod from falling. The Lunar Lander agent has to land a spacecraft on the moon, controlling three thrusters to avoid crashing. In Minigrid Door Key, the agent needs to explore to find a key,

unlock the door, and reach the goal. This is a partially observable domain, and the shaded area means the agent's current observable area.

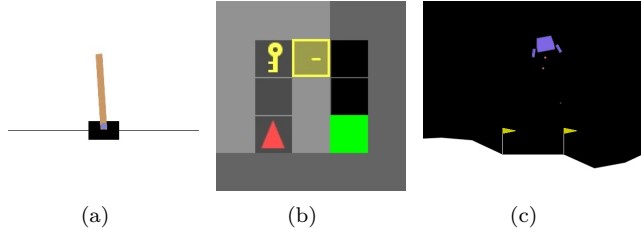

(a)  (b)  (c)

Figure 2: (a) Cart Pole, (b) Minigrid Door Key, and (c) Lunar Lander are used for evaluation

**Evaluation metrics:** The agent is evaluated based on its accumulated reward on evaluation episodes by executing the current policy. During evaluation episodes, the agent's policy is frozen, hence no exploration. Recall that the agent does not have access to the reward signal — however, we use the default built-in reward function of our domains to evaluate the agent performance. We also use the % of correct feedback of the filtered data, namely the *pure ratio* to evaluate the algorithm's noise-filtering capability.

**Experimental Settings:** The feedback is provided by a scripted teacher. The ultimate goal of this algorithm is to learn from noisy human feedback, and a fixed teacher allows for better evaluation rigor and repeatability. The teacher uses a pre-trained expert policy for each domain, providing negative feedback when the agent fails to choose the optimal action, and positive feedback otherwise. In our settings, just like real human teachers who cannot provide feedback at every step, the agent receives feedback at fixed intervals of time steps. § The feedback's noise is symmetric, i.e., all feedback labels are randomly and independently flipped by a fixed probability depending on the noise ratio, which is less than 50%. The maximum number of feedback that an agent can receive is defined as its budget. CANDERE-COACH has a pretrained classifier that uses a small non-noisy feedback dataset and also has access to a limited number of noisy feedback (budget).

**Baselines:** The two baselines that we compare CANDERE-COACH against are: (1) Deep COACH, which is allowed the same amount of noise-free feedback budget at the beginning of an episode as CANDERE-COACH; this budget is the same as the pretraining dataset of our algorithm, but consists of random states and actions that the agent visited (2) Deep COACH (Preload), which loads the exact same dataset for classifier pretraining into the replay buffer as CANDERE-COACH. These existing methods assume perfect teacher feedback with no mechanism to handle corrupted signals, making them the natural baselines for isolating the contribution of our noise-aware framework. We query feedback every 10 steps in a limited budget setting to avoid sampling redundant feedback labels from similar states. In Cart Pole, the budget is 1000 and it depletes at 10000 iterations. In MiniGrid Door Key, the budget is 500 and it depletes at 5000 iterations. In Lunar Lander, the domain is relatively complicated with a continuous state space and we have the budget of 5000 which depletes at 50000 iterations. Noticeably, PEBBLE (Lee et al., 2021) and other methods (Cheng et al., 2024; Huang et al., 2025) to identify noisy preferences do not serve as a baseline since it requires preferences over trajectories, while in our settings, the teacher provides feedback over a set of states and actions. In our experiments, the baselines always have the same number of feedback budget to ensure a fair comparison. Each experiment consists of 10 runs of different seeds. ¶

## 5.1 Results

**When is Deep COACH sensitive to noisy feedback?** To answer **RQ1**, we conduct experiments to evaluate Deep COACH with different proportions of noise. We evaluated Deep COACH with noise amounts of {0%, 10%, 20%, 30%, 40%} in the three domains. We consider both limited and unlimited feedback settings for this experiment. As shown in Figure 3(a) and Figure 3(b) for Cart Pole, we observe that higher noise leads to worse agent performance (less reward) and/or more time required to converge to optimal

---

§The details of feedback frequency can be found in Appendix Section H.

¶More experimental settings and hyperparameter details can be found in Appendix Sections H.

performance. Interestingly, with an unlimited budget, the agent is still able to learn with 40% feedback noise. Statistically, by the law of large numbers, as long as there are more correct labels than incorrect ones and unlimited feedback data, the agent eventually learns the correct policy given infinite amounts of feedback and learning time. However, for limited budget feedback, as shown in Figure 3(b), the agent performance significantly deteriorates and even unlearns over time. Similar results can also be found in the other two domains (Lunar Lander and Minigrid Door Key), shown in Appendix Section A.

To summarize, we address **RQ1** by showing that noisy feedback poses a significant negative impact on Deep COACH, especially in a *limited feedback setting.*

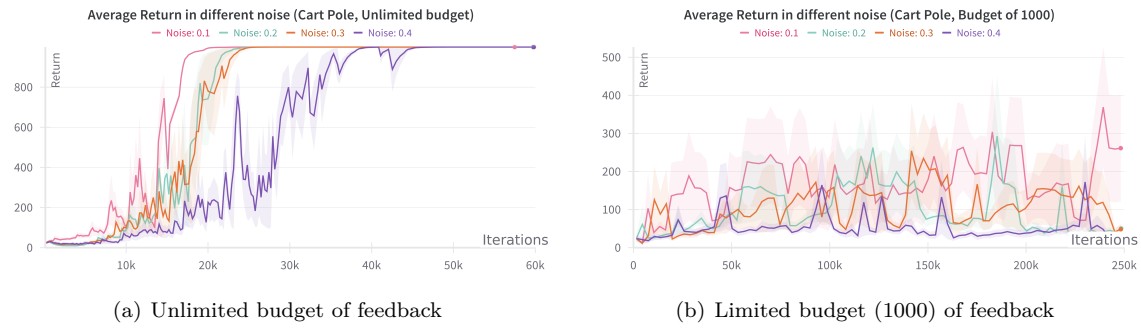

(a) Unlimited budget of feedback        (b) Limited budget (1000) of feedback

Figure 3: Performance of Deep COACH under different scales of noises in Cart Pole. While with an unlimited budget the Deep COACH is even able to learn against 40% noise slowly, the performance of Deep COACH significantly deteriorates with a limited budget.

**CANDERE-COACH evaluation:** To answer **RQ2** and **RQ3**, we evaluate the performance of the original CANDERE-COACH with a limited feedback budget based on the performance of Deep COACH from the previous experiments. We also evaluate CANDERE-COACH without active relabelling, denoted as CANDERE-COACH (w/o AR). In the figures, the x-axis denotes learning iterations. Before budget depletion, each iteration corresponds to one environment interaction step (and one model update step). After budget depletion, iterations correspond only to model update steps; y-axis denotes the average evaluation return.

**Cart Pole:** As shown in Figure 4(a), CANDERE-COACH, can learn well (higher return) with 30% noise in *Cart Pole* and outperform Deep COACH, which shows highly unstable performance. CANDERE-COACH (w/o AR) is also able to outperform Deep COACH in performance, however, it cannot match CANDERE-COACH in performance which shows that active-relabeling for noise correction helps in identifying clean feedback from suspected noisy feedback. With 40% noise (shown in Figure 4(d)), CANDERE-COACH learns well, but the performance is unstable due to high noise. However, it still outperforms CANDERE-COACH (w/o AR) in terms of average return towards the end of learning. Deep COACH fails to learn and receives low episodic rewards and a similar pattern is observed for Deep COACH (Preload).

**Door Key:** As shown in Figure 4(b), CANDERE-COACH is able to outperform Deep COACH and Deep COACH (Preload) under 30% noise. CANDERE-COACH (w/o AR) does not perform well with almost the same performance as the baselines. When the noise scale increases to 40%, CANDERE-COACH (w/o AR) cannot perform well and only exceeds Deep COACH slightly in performance; though not statistically significant. The best performance is achieved by CANDERE-COACH as shown in Figure 4(e), surpassing the baselines.

**Lunar Lander:** As suggested by Figure 4(c), CANDERE-COACH (w/o AR) is not able to learn in this domain. Although both baselines fail to achieve satisfactory performance, CANDERE-COACH (w/o AR) is impacted even more by noise, due to its classifier also receiving negative influence from noise during training and further deteriorates the learning ability of the agent. CANDERE-COACH however, shows superior performance against 30% noise, outperforming all the baselines that fail to reach an episodic return of 200.‖ However, Lunar Lander is still challenging under extremely high noise (40%) — as shown in Figure 4(f),

---

‖In Lunar Lander, only an episodic return over 200 is considered a success.

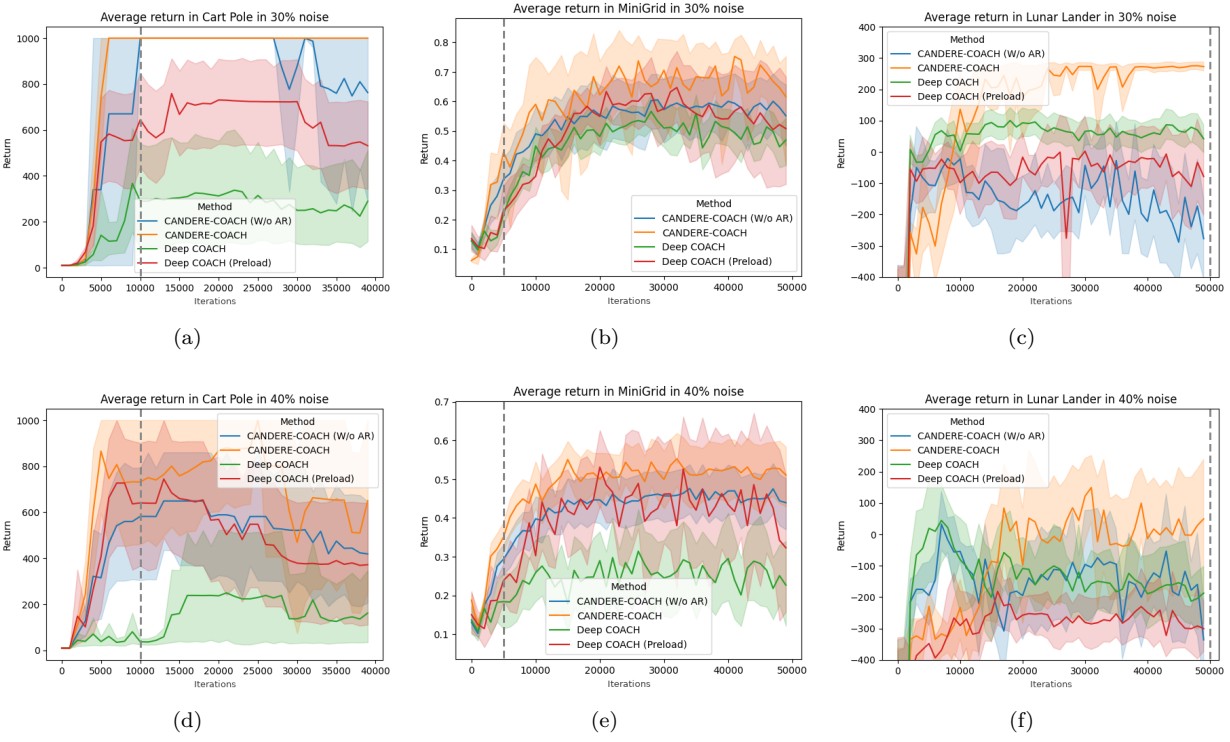

Figure 4: Performance of CANDERE-COACH in Cart Pole, Door Key and Lunar Lander in 30% and 40% noise. The x-axis is learning iterations for the policy and the classifier. The dashed grey line denotes the budget depletion.

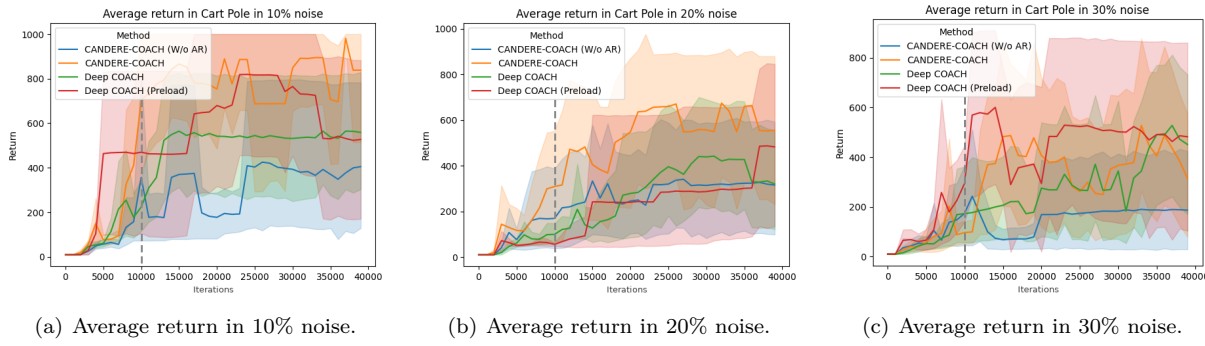

(a) Average return in 10% noise.  (b) Average return in 20% noise.  (c) Average return in 30% noise.

Figure 5: Performance of CANDERE-COACH in Cart Pole, with a noisy pretraining dataset. The x-axis is learning iterations for the policy and the classifier. The dashed grey line denotes the budget depletion.

CANDERE-COACH still achieves the best episodic return as compared to the baselines, but it is impacted by noise and fails to get an average return of 200 at the end.

In summary, CANDERE-COACH is generally effective in filtering noise compared to other baselines. Its performance on average is better with relatively less noise (up to 30%), as compared to very high noise (40%); thus affirmatively answering RQ2. CANDERE-COACH with active relabeling achieves the strongest performance across domains and noise levels as compared to all the other algorithms, with statistically significant gains in most settings, thus supporting RQ3. **

---

** For the performance of CANDERE-COACH with other noise levels, please refer to Section D of the Appendix.

## 5.2 Ablation studies

We conduct additional ablation studies to understand the components and effects of hyperparameters on the performance of CANDERE-COACH.

**Effect of online training on noise-filtering classifier:** Online training of the classifier tries to balance two problems. First, the state action distribution can shift (relative to the data used to pretrain the classifier), suggesting online training will help. Second, trying to update the classifier with noisy labels could hurt performance, suggesting online training will not help. We denote the CANDERE-COACH without online training and active relabelling as CANDERE-COACH (w/o AR, w/o OT). Figure 6(a) shows that for CANDERE-COACH (w/o AR, w/o OT), the pure ratio reduces over time, suggesting that as the agent explores different areas of the state-action space, the distribution shift leads to worse classifier performance. In contrast, online training allows the pure ratio to gradually increase. Eventually, the pure ratio stabilizes around 95% and as a result, CANDERE-COACH (w/o AR) is able to learn robustly against 30% noise with pretraining dataset of size 25, shown in Figure 6(b).

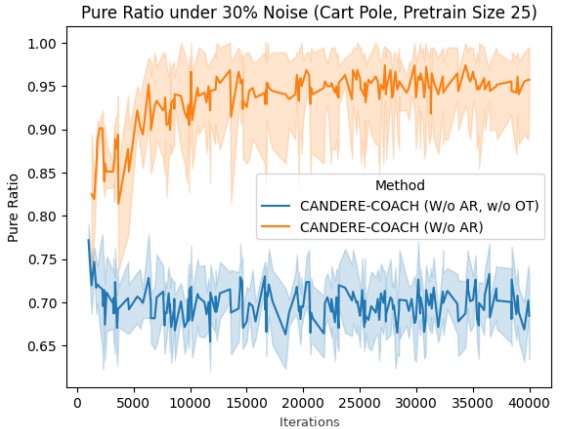

(a) Pure ratio with and without online training in Cart Pole under 30% noise.

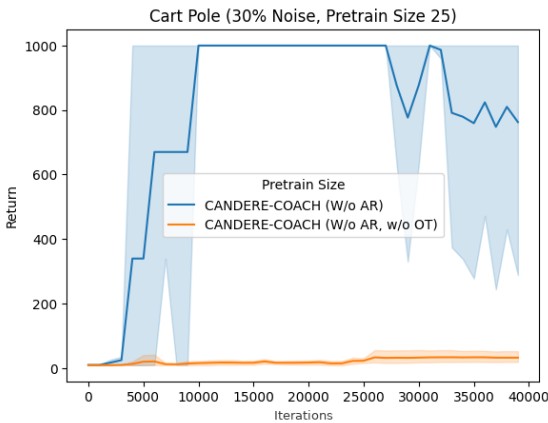

(b) Average return with and without online training in Cart Pole under 30% noise.

Figure 6: Effect of online training in Cart Pole under 30% noise. As the agent explores new states and actions, the state-action distribution shifts, so a fixed classifier becomes less accurate over time without online training. With online learning, CANDERE-COACH (w/o AR) succeeds using only a pretraining dataset of size 25.

Yet, online training does not always show such a great improvement when the noise level increases. The results with 40% noise can be found in Figure 7. We also observe that the pure ratio under extremely high noise is reduced and unstable, which differs from the pattern seen in 30% noise, which shows that the online training mechanism tends to perform worse under very high noise. However, its performance remains better than CANDERE-COACH without online training both in average return (shown in Figure 7(b)) and pure ratio (shown in Figure 7(a)).

To summarise, with online training, CANDERE-COACH has a better sample efficiency and leads to a higher pure ratio during training.

**Effect of noisy pretraining dataset:** In prior sections, CANDERE-COACH was allowed access to a small noise-free feedback dataset for pretraining the classifier. In this section, we invalidate this assumption by testing CANDERE-COACH with a noisy pretraining dataset. As observed in Figure 5, CANDERE-COACH can still perform well under low amounts of noise, such as 10% and 20% (see Figures 5(a) and Figure 5(b)), while CANDERE-COACH (w/o AR) cannot outperform our baseline. However, if the noise level is too high (30%), as shown in Figure 5(c), we observe that the performance of CANDERE-COACH begins to degrade. Also, pretraining with a noisy dataset reduces the overall performance compared to the performance reported

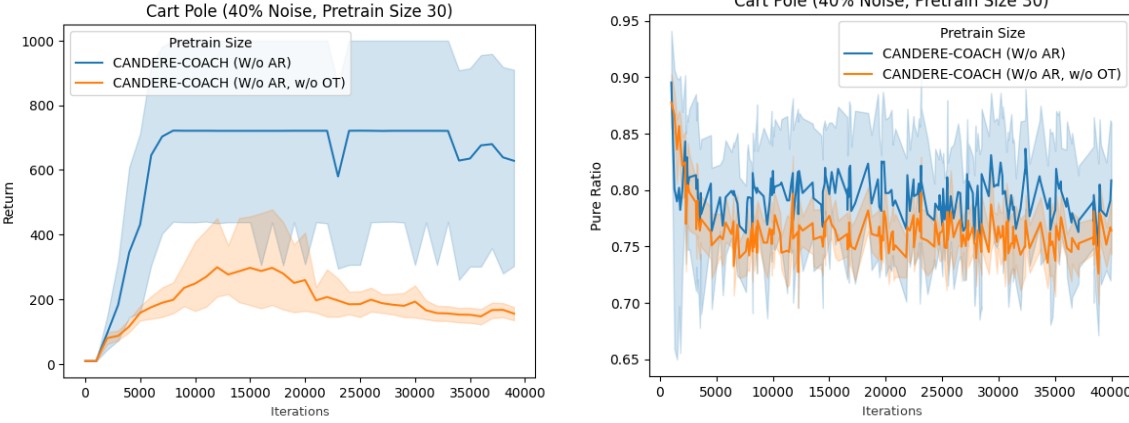

(a) Average return with and without online training in Cart Pole in 40% noise.

(b) Pure ratio with and without online training in Cart Pole in 40% noise.

Figure 7: Ablation study on online training: average episode return and pure ratio of CANDERE-COACH (with and without OT) and Deep COACH in Cart Pole under 40% noise.

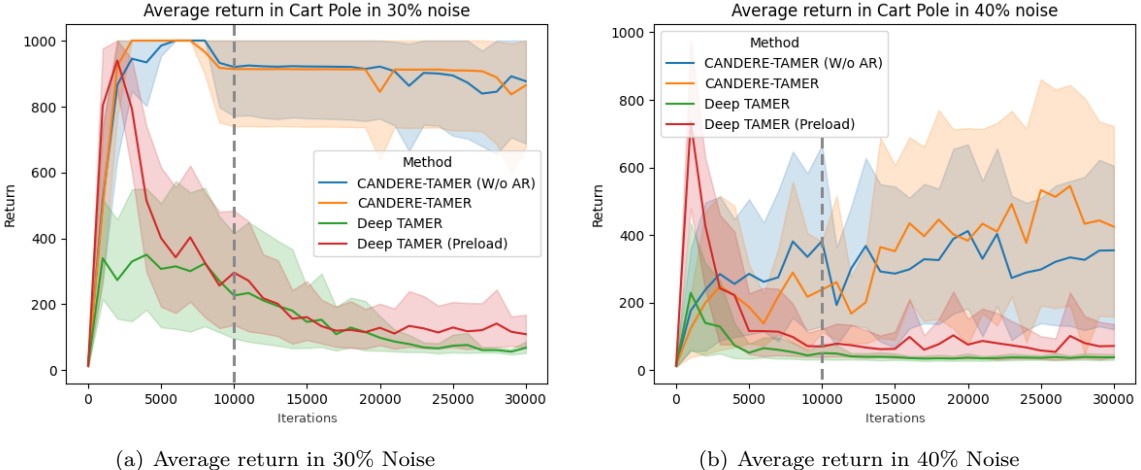

(a) Average return in 30% Noise

(b) Average return in 40% Noise

Figure 8: Performance comparison of CANDERE-TAMER in Cart Pole.The dashed grey line denotes the budget depletion

in previous sections, where a noise-free pretraining dataset is available. To summarise, CANDERE-COACH can work with a noisy pretraining dataset but it only shows promising results under reasonably low noise levels like 10% or 20% noise.

## 5.3 Extending to CANDERE-TAMER

There is no fundamental challenge to expand our proposed de-noising mechanism to other learning from feedback algorithms. Here, we present CANDERE-TAMER, which is similar to CANDERE-COACH but built based on Deep TAMER (Warnell et al., 2018). The results are shown in Figure 8. We observe a similar pattern in which our algorithms outperform our baselines (Deep TAMER and Deep TAMER (Preload)) and learn successfully against 30% noise, while the performance of baselines degrades significantly with noise. In 40% noise, CANDERE-TAMER still shows a better average return and outperforms our baselines; however, its performance is significantly worse than that with 30% noise. This experiment shows the potential of

our approach to be used as a plug and play tool inside any human-in-the-loop RL algorithm, which will be explored as future work.

## 6 Conclusion and Future Work

In this work, a new algorithm is proposed inside the learning from feedback RL framework. Through experiments in multiple settings and tasks, we show that the CANDERE-COACH is able to handle up to 40% noise, with a small noise-free feedback dataset, outperforming the baselines. We also show that if the noise is small enough, our CANDERE-COACH can also work with a noisy pretraining dataset. Besides Deep COACH, we further show that the proposed noise detection mechanism can be extended to Deep TAMER. A limitation of our proposed work is that it's restricted to domains with discrete actions, as collecting state–action level binary feedback is more realistic in these settings. Our structure enables the teacher to provide meaningful and actionable feedback, in contrast to continuous control tasks where such feedback is significantly difficult to obtain.

For future work, we intend to expand this framework to learning from preferences algorithms and also test its performance with different types of noise, such as non-symmetric and feature-dependent noise, as well as conducting a human subject study.

## 7 Acknowledgement

Part of this work has taken place in the Intelligent Robot Learning (IRL) Lab at the University of Alberta, which is supported in part by research grants from the Alberta Machine Intelligence Institute (Amii); a Canada CIFAR AI Chair, Amii; Digital Research Alliance of Canada; and the National Science and Engineering Research Council (NSERC).

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

## A   Deep COACH with noise

We also conducted experiments to study the impact of noise on our baseline (Deep COACH) in other domains. In Door Key, noticeably, the performance of COACH seems less impacted with a limited budget, with a smaller average return drop, shown in Figure 9. This is mainly due to the unique settings in Door Key. In Door Key, the agent needs to grab the key, unlock the door, and reach the destination. An evaluative reward of positive +1 discounted by time steps is given when it reaches the destination. Even a randomly initialised agent can reach the destination and achieve a nonzero return at step 0, which is different from Cart Pole and Lunar Lander, where a bad policy results in low episodic returns. Therefore, the performance in Door Key seems to receive less negative impact with a limited budget. However, we still observe a drop in performance. For example, in 40% noise, with an unlimited feedback budget, the return reaches up to 0.4 after 50k steps, while with a limited budget it converges around 0.2.

For Lunar Lander, we observe the same trends as Cart Pole, as shown in Figure 10; the agent's performance for the unlimited feedback setting decreases as noise % increases. However, the agent is still able to learn, though not as good as learning from clean feedback (0% noise). With limited feedback, the agent's learning performance deteriorates significantly, and the agent eventually unlearns over time.

## B   CANDERE-COACH with unlimited budget

In this section, we evaluate the performance of the original CANDERE-COACH with an unlimited budget. We conducted experiments with different sizes of pretraining datasets for the classifier.

As can be seen from Figure 11, CANDERE-COACH is able to outperform when we have a pretraining data size of 30, against 40% of noise. Though CANDERE is able to learn faster than the baseline, with an unlimited budget of feedback, both the COACH and CANDERE is able to learn and reach almost perfect performance eventually. Furthermore, when the classifier does not have enough data for pretraining and cannot select correct labels for agent updates, the performance will be downgraded to a very low level. Here, to conclude, with a small amount of feedback dataset, CANDERE-COACH is able to outperform Deep COACH by learning faster. But with an unlimited budget of feedback, even our baseline is able to learn well against high noise.

## C   Ablation study on pretraining sizes

As shown in Figure 12, our method can learn well against 30% noise, while Deep COACH shows highly unstable performance. In fact, even with different amounts of preloaded pure dataset, under the same settings, Deep COACH shows a significantly different learning curve due to the very unpredictability of the noise. Furthermore, it is revealed in Figure 13 that a classifier pretrained with datset of size 25 barely filters noise successfully as the pure ratio hovers around 70% under 30% noise, while the other two classifiers successfully improve the pure ratio up to 80% and 85%. We also tested CANDERE-COACH on 40% noise,

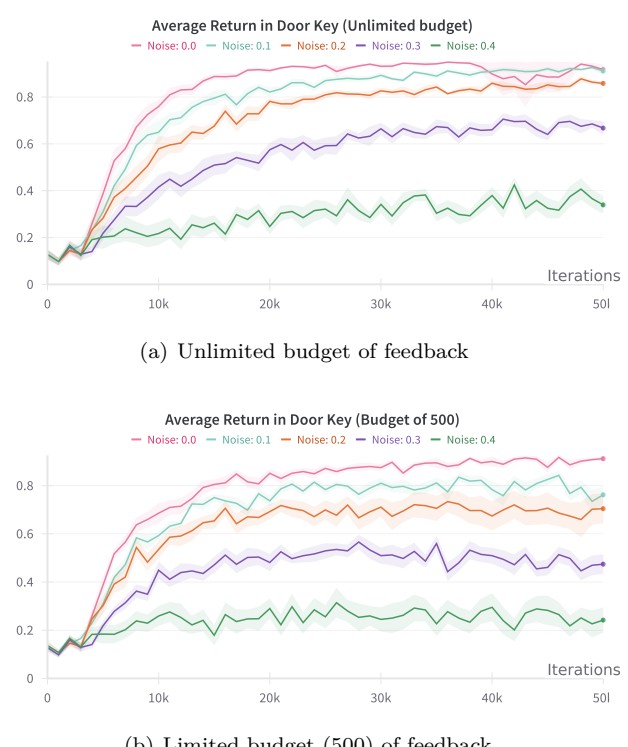

(a) Unlimited budget of feedback

(b) Limited budget (500) of feedback

Figure 9: Performance of Deep COACH under different scales of noises in Door Key

as shown in Figure 14. With extremely high noises, CANDERE-COACH requires more pretraining data (35 in both 40%) to succeed. We can also observe a similar pattern in pure ratio, which keeps decreasing and then becomes stable.

The results of CANDERE-COACH (w/o AR) in 30% noise can be found in Figure 15. Recall that in the previous section, CANDERE-COACH fails to learn with a pretraining dataset of size 25. With online training, we observe that the pure ratio can gradually increase, which means that the classifier is learning and is well adapted to the new state and action distribution. Eventually, the pure ratio stabilises around 95% and as a result, CANDERE-COACH with online training is able to learn robustly against 30% noise with a pretraining dataset of size 25. However, online training does not always show such a great improvement when noise increases. The results in 40% noise can be found in Figure 16. Under 40% noise, CANDERE-COACH with online training fails with 25 pretraining data and needs 30 to succeed. We also observe that the pure ratio under extremely high noise will go down and then fluctuate, which differs from the pattern in Figure 15 and Figure 13, which shows that the online training mechanism tends to perform worse under high noise.

## D   CANDERE-COACH in other noise levels

We also conduct experiments in lower noise level like 20%, the results are shown in Figure 17. With lower noise, in Cart Pole and Door Key, the performance difference is less significant because our baseline Deep COACH (Preload) can also perform well. However, Lunar Lander is our hardest domain, and we still observe a similar pattern in 20% noise, where CANDERE-COACH shows the best performance and reaches 200 in episodic average return, while CANDERE-COACH (w/o AR) and baselines fail to do so.

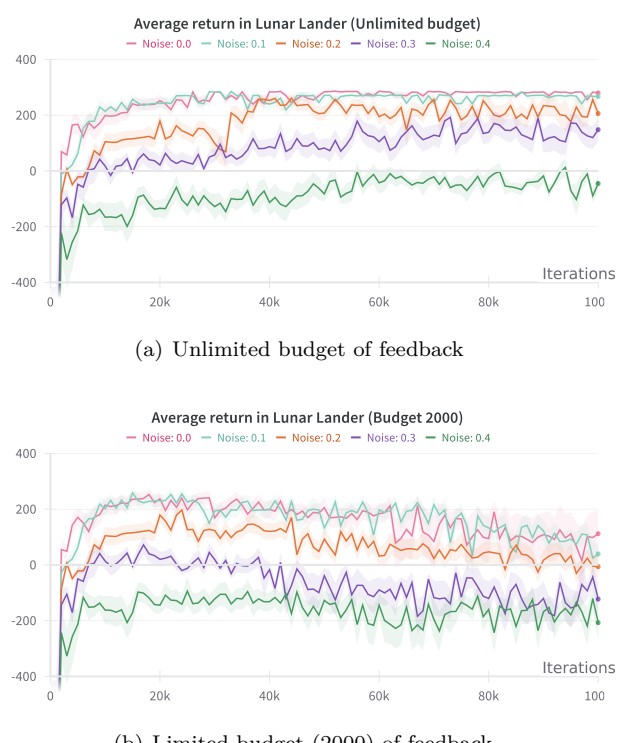

(a) Unlimited budget of feedback

(b) Limited budget (2000) of feedback

Figure 10: Performance of Deep COACH under different scales of noises in Lunar Lander

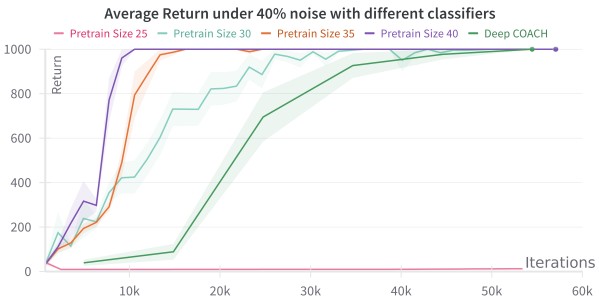

Figure 11: Performance of CANDERE-COACH under 40% noise, with different amounts of pretraining data size.

# E  Study on feedback budget

We conduct most of our experiments with a limited budget. As different budgets can lead to different results, the budget for experiments is chosen based on such criteria: the budget should at least allow the agent to solve the task with 0% noise, while the agent may fail with higher noise levels.

A study on the budget's influence on Deep COACH with a noise-free budget of 30 is shown in Figure 18. A budget of 1000 allows the agent to reach very close to maximum episodic return smoothly, while the performance drops as noise increases. With a smaller budget like 750, the agent fails to do so and hence we choose 1000 as the budget for Cart Pole in our experiments for a comparison between Deep COACH and CANDERE-COACH.

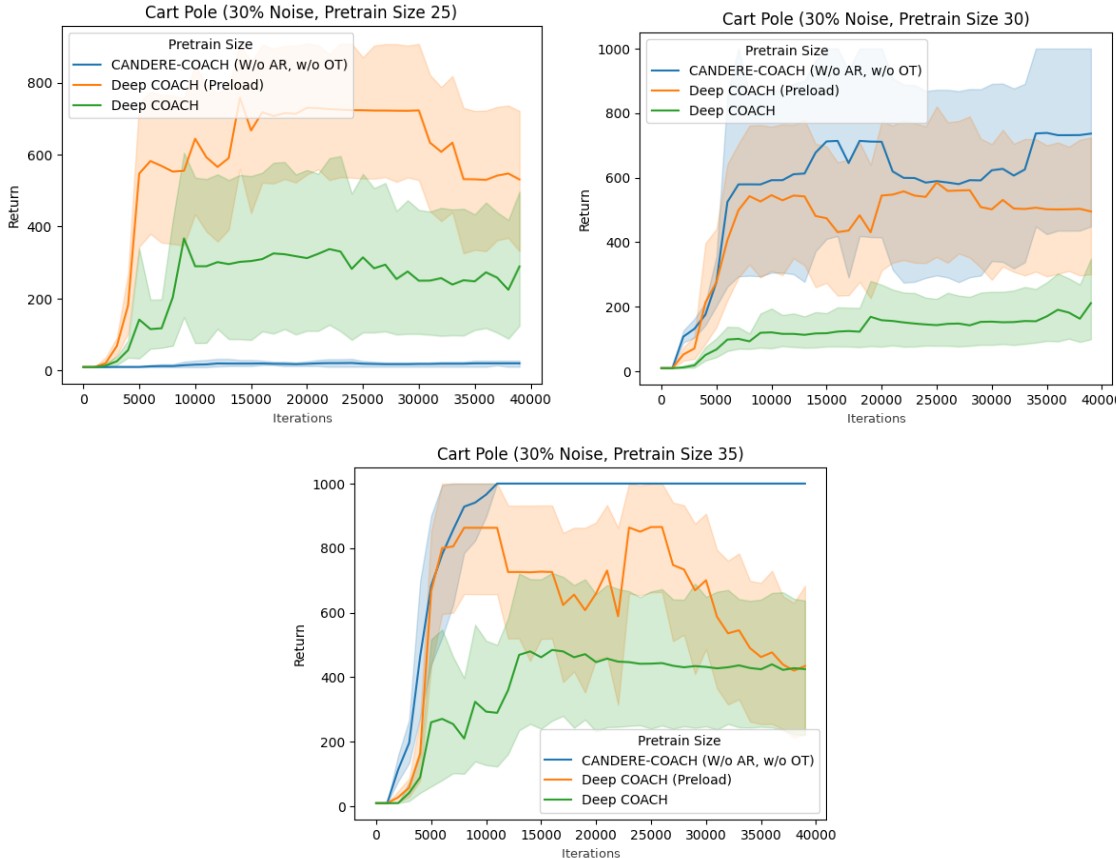

Figure 12: Ablation study on pretraining sizes: average episode return of CANDERE-COACH and Deep COACH in Cart Pole under 30% noise. Three figures show the same experiment with different amounts of pretraining feedback. It can be seen that CANDERE-COACH needs at least 30 to succeed and 35 to perform well.

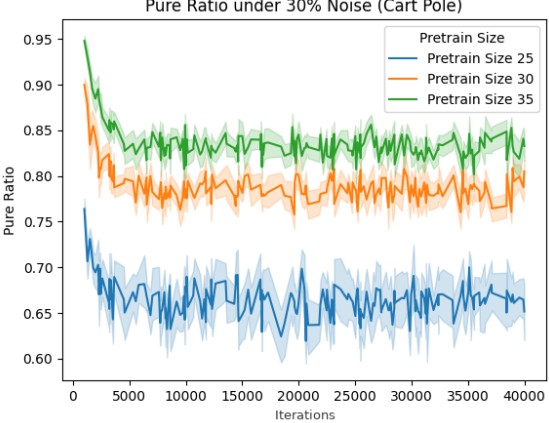

Figure 13: Ablation study on pretraining sizes: average pure ratio of CANDERE-COACH in Cart Pole under 30% noise. While the agent explores new states and actions, the distribution of state and action changes and therefore a fixed classifier predicts less accurately over time.

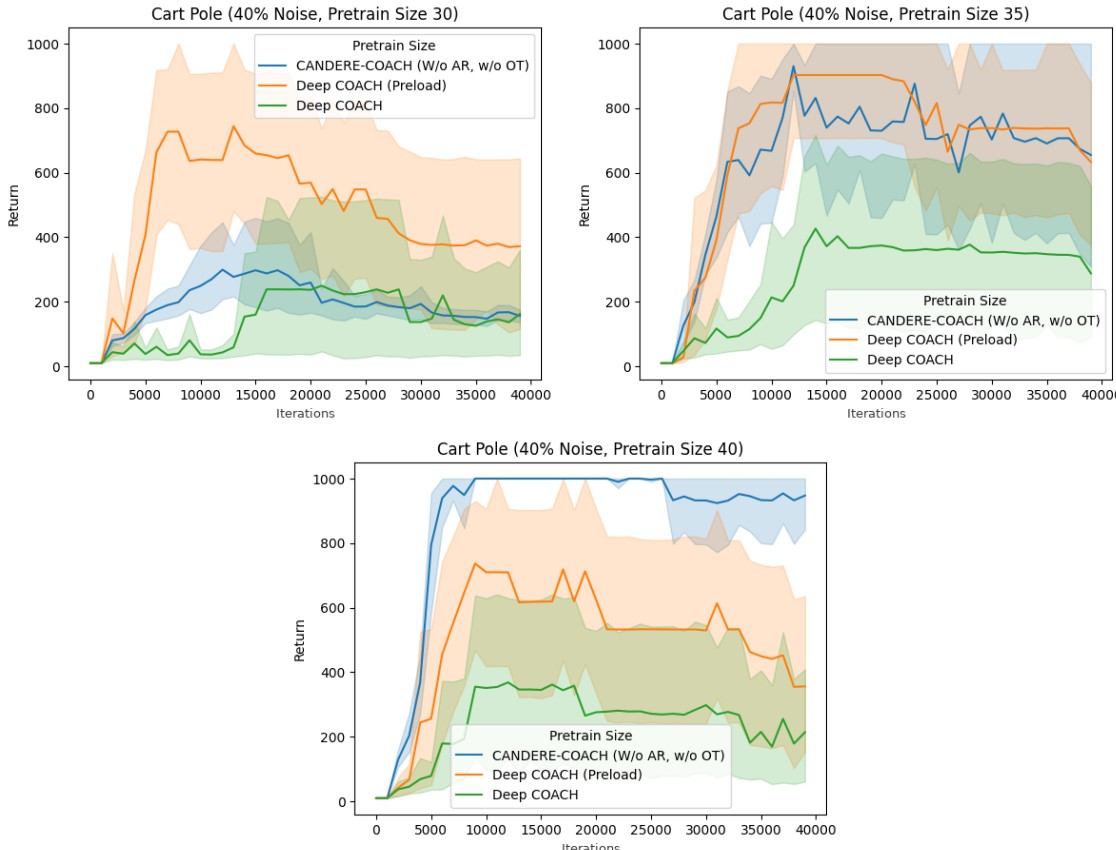

Figure 14: Ablation study on pretraining sizes: average episode return of CANDERE-COACH and Deep COACH in Cart Pole under 40% noise. Three figures show the same experiment with different amounts of pretraining feedback. 40% is much harder and our CANDERE-COACH will also suffer from noise with pretraining size of 30 and CANDERE-COACH needs 40 to reach the maximum episodic return (1000).

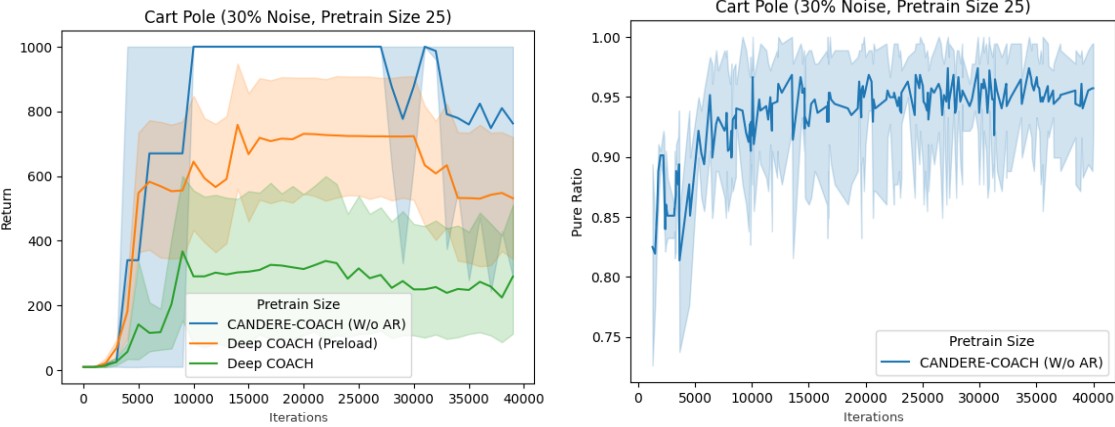

Figure 15: Performance of CANDERE-COACH with online training in 30% noise. With online training, CANDERE-COACH is able to learn against 30% noise with merely a pretraining dataset of size 25. Furthermore, its pure ratio successfully increases over time and reaches 95%, while CANDERE-COACH without online training decreases over time.

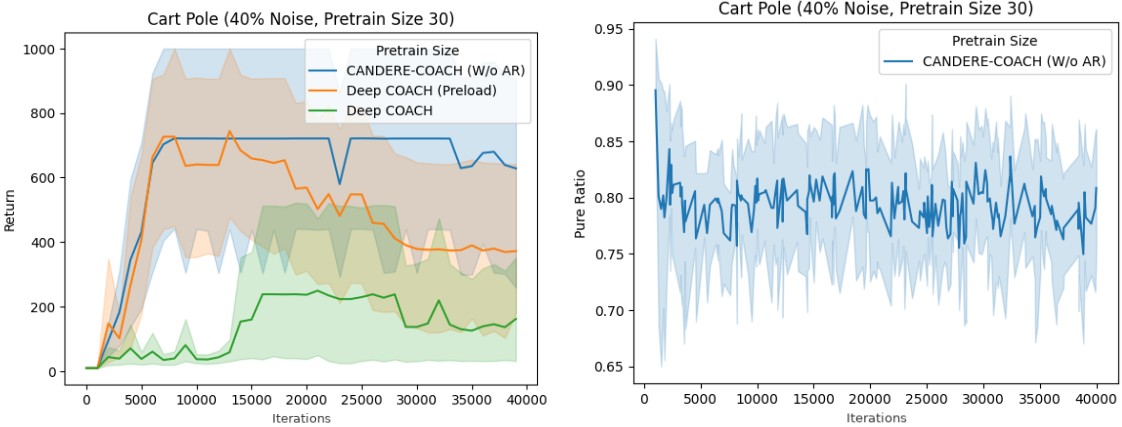

Figure 16: Performance of CANDERE-COACH with online training in 40% noise.

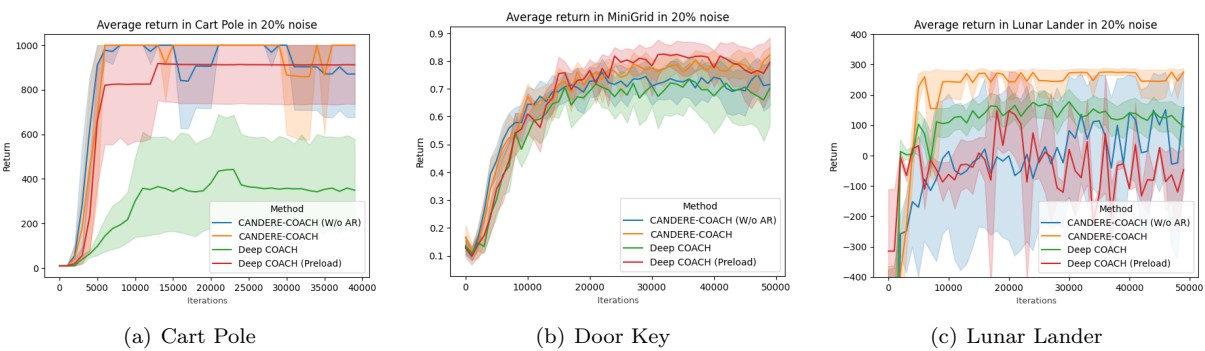

(a) Cart Pole        (b) Door Key        (c) Lunar Lander

Figure 17: Performance comparison in 20% noise in Cart Pole, Door Key and Lunar Lander

## F  Scripted teacher

Our scripted teacher is trained with PPO Schulman et al. (2017) following the hyperparameters of RL Zoo Raffin (2020). More details are described in the following subsections for each domain.

### F.1  Cart Pole

In Cart Pole, the expert is trained with PPO with hyperparameters shown in Table 1.

### F.2  Door Key

In Door Key, the expert is trained with PPO with hyperparameters shown in Table 2. Furthermore, Minigrid Door Key is set to be fully observable and we use a CNN-based feature extractor to reduce the observation space to 5.

### F.3  Lunar Lander

In Lunar Lander, the expert is trained with PPO with hyperparameters shown in Table 3.

| Hyperparameter | Value |
|---|---|
| Time steps | 1e5 |
| Rollout steps | 32 |
| Gae lambda | 0.8 |
| Gamma | 0.9 |
| Learning rate | 1e-3 |
| Clip range | 0.2 |
| Batch size | 256 |
| Hidden units | 64 |
| Layers | 2 |
| Activation | ReLU |

Table 1: Hyperparameters of Cart Pole expert training

| Hyperparameter | Value |
|---|---|
| Time steps | 1e5 |
| Rollout steps | 128 |
| Gae lambda | 0.95 |
| Gamma | 0.99 |
| Learning rate | 2.5e-4 |
| Clip range | 0.2 |
| Batch size | 64 |
| Hidden units | 64 |
| Layers | 2 |
| Activation function | ReLU |
| CNN channels | $[16, 32, 64]$ |
| CNN kernel size | $2, 2$ |

Table 2: Hyperparameters of Door Key expert training

| Hyperparameter | Value |
|---|---|
| Time steps | 1e6 |
| Rollout steps | 1024 |
| Gae lambda | 0.98 |
| Gamma | 0.999 |
| Learning rate | 1e-3 |
| Clip range | 0.2 |
| Batch size | 64 |
| Hidden units | 64 |
| Layers | 2 |
| Activation | ReLU |

Table 3: Hyperparameters of Lunar Lander expert training

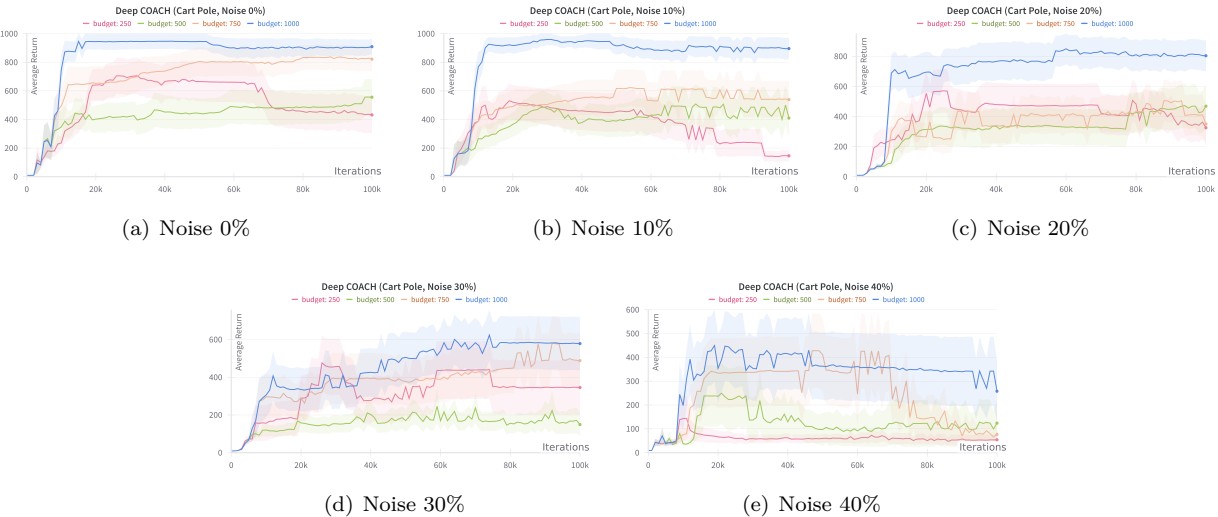

$$\text{(a) Noise 0\%} \qquad \text{(b) Noise 10\%} \qquad \text{(c) Noise 20\%}$$

$$\text{(d) Noise 30\%} \qquad \text{(e) Noise 40\%}$$

Figure 18: Performance of Deep COACH (Preload) under different scales of noise and budget in Cart Pole

## G  Collecting the pretraining dataset

The pretraining dataset is collected with our scripted teacher to label with state-action pairs, positive or negative feedback. We first collect states following a certain distribution to ensure better coverage of the state space. Then we label the states and optimal actions to be positive, and the collected dataset will be randomly shuffled. Lastly, we practice data augmentation by labelling all the non-optimal actions to be negative. The details of the state sampling distribution of each domain are described in the following subsections.

### G.1  Cart Pole

In Cart Pole, the state is sampled uniformly in a clipped observation space. The state space of Cart Pole consists of four dimensions and two of them are not bounded. Therefore, we properly choose a suitable clip range to sample the states. We set the sampling range of position and velocity based on the fact that an episode will be terminated if the cart leaves the $(-2.4, 2.4)$ range. Details can be seen in Table 4.

| Dimension | Original Min | Clipped Min | Original Max | Clipped Max |
|---|---|---|---|---|
| Cart Position | -4.8 | -2.4 | 4.8 | 2.4 |
| Cart Velocity | -Inf | -2.4 | +Inf | 2.4 |
| Pole Angle | -0.418 | -0.418 | 0.418 | 0.418 |
| Pole Angular Velocity | -Inf | -0.418 | +Inf | 0.418 |

Table 4: Sampling space of Cart Pole

### G.2  Door Key & Lunar Lander

In Door Key and Lunar Lander, the dataset is sampled from trajectories following a sampling policy that takes expert action by 50% chance and random action by 50% chance. The reason for this is different for these domains. In Door Key, we cannot practice uniform sampling in the RGB image array space. In Lunar Lander, the observation space is significantly larger and if we practice uniform sampling, most sampled states will never be visited by the agent and therefore bring low performance of the classifier.

## H  COACH Hyperparameters

In this section, we show the hyperparameters used in our experiments for CANDERE-COACH and Deep COACH. Deep COACH shares the same hyperparameters with CANDERE-COACH if applicable.

### H.1  Cart Pole

The hyperparameters in Cart Pole can be seen in Table 5.

| Hyperparameter | Value |
|---|---|
| Actor learning rate | 0.00005 |
| Batch size | 256 |
| Budget | 1000 |
| Eligibility trace windows size | 10 |
| Eligibility trace decay factor | 0.35 |
| Classifier learning rate | 0.01 |
| Feedback frequency | 10 |
| Actor hidden units | 1024 |
| Actor layers | 2 |
| Actor activation | ReLU |
| Q function hidden units | 1024 |
| Q function layers | 2 |
| Q function activation | ReLU |
| Classifier hidden units | 64 |
| Classifier layers | 2 |
| Classifier activation | ReLU |
| Classifier pretraining epochs | 100 |
| Classifier pretraining learning rate | 0.001 |
| Classifier pretraining loss | Cross entropy loss |
| Active relabelling rate | 0.6 |

Table 5: Hyperparameters of CANDERE-COACH in Cart Pole

### H.2  Door Key

The hyperparameters in Door Key can be seen in Table 6.

### H.3  Lunar Lander

The hyperparameters in Lunar Lander can be seen in Table 7.

### H.4  Summary on variations of CANDERE-COACH

We summarise the aforementioned CANDERE-COACH and its variations' domain-wise performance and required amount of pretraining dataset size, as shown in Table 8.

## I  Study on Other Algorithm Variants

We additionally conducted ablation study based on two different criteria: (1) Uncertainty-based estimates (2) Labelling agent transitions using the denoising-based classifier.

**1. Ablation based on uncertainty based estimates**. The variants considered within this ablation are (1) **Co-Teaching**: classifier start from scratch, following the previous work in Han et al. (2018) and use

| Hyperparameter | Value |
| --- | --- |
| Actor learning rate | 0.00005 |
| Batch size | 256 |
| Budget | 500 |
| Eligibility trace windows size | 10 |
| Eligibility trace decay factor | 0.35 |
| Classifier learning rate | 0.001 |
| Feedback frequency | 10 |
| Actor hidden units | 1024 |
| Actor layers | 2 |
| Actor activation | ReLU |
| Q function hidden units | 1024 |
| Q function layers | 2 |
| Q function activation | ReLU |
| Classifier hidden units | 64 |
| Classifier layers | 2 |
| Classifier activation | ReLU |
| Classifier pretraining epochs | 100 |
| Classifier pretraining learning rate | 0.001 |
| Classifier pretraining loss | Focal loss |
| Active relabelling rate | 0.8 |

Table 6: Hyperparameters of CANDERE-COACH in Door Key

| Hyperparameter | Value |
| --- | --- |
| Actor learning rate | 0.00005 |
| Batch size | 256 |
| Budget | 5000 |
| Eligibility trace windows size | 10 |
| Eligibility trace decay factor | 0.35 |
| Classifier learning rate | 0.001 |
| Feedback frequency | 10 |
| Actor hidden units | 1024 |
| Actor layers | 2 |
| Actor activation | ReLU |
| Q function hidden units | 1024 |
| Q function layers | 2 |
| Q function activation | ReLU |
| Classifier hidden units | 64 |
| Classifier layers | 2 |
| Classifier activation | ReLU |
| Classifier pretraining epochs | 100 |
| Classifier pretraining learning rate | 0.001 |
| Classifier pretraining loss | Focal loss |
| Active relabelling rate | 0.6 |

Table 7: Hyperparameters of CANDERE-COACH in Lunar Lander

| Algorithm
Domain &Noise | CANDERE-COACH (w/o AR, w/o OT) | CANDERE-COACH (w/o AR) | CANDERE-COACH |
|---|---|---|---|
| Cart Pole, 30% | Outperform with pretrain size 30 | Outperform with pretrain size 25 | Outperform with pretrain size 20 |
| Cart Pole, 40% | Outperform with pretrain size 30 | Outperform with pretrain size 30 | Outperform with pretrain size 25 |
| Door Key, 30% | Fail with pretrain size 12 | Outperform with pretrain size 12 | Outperform with pretrain size 12 |
| Door Key, 40% | Fail with pretrain size 12 | Fail with pretrain size 12 | Outperform with pretrain size 12 |
| Lunar Lander, 30% | Fail with pretrain size 150 | Fail with pretrain size 150 | Outperform with pretrain size 150 |
| Lunar Lander, 40% | Fail with pretrain size 150 | Fail with pretrain size 150 | Outperform with pretrain size 150 |

Table 8: Summary of results of CANDERE-COACH and its variations

the small loss trick for denoising; (2) **Uncertainty**: initialise an ensemble of pretrained classifiers (size=5) and use the variance to predict clean or noisy labels; (3) **Blend**: similar to uncertainty but uses a 50%-50% mixed score of ensemble variance and the small loss trick to denoise and (4) **Agreement**: instead of using loss as an indicator, learn from samples where the classifier predicts the same as the received label (i.e., the classifier agrees with the label). In order to ensure a fair comparison, all experiments use the same pretrained classifier trained on the same dataset. For the ensemble-based methods (Uncertainty, Blend), each classifier in the ensemble is initialized from the same pretrained weights, and the ensemble models subsequently diverge during online training due to different sampled mini-batches.

The results are shown in Figure 19, 20 and 21. As we can see, CANDERE-COACH generally stays stable and outperforms the other variants except for Minigrid where the Agreement version jumpstarts faster than CANDERE-COACH for all noise levels. While other de-noise mechanism are often worse, we find that agreement is a very conservative strategy as it only picks the labels that the classifier agrees on, and therefore its performance is dependent on the initial classifier and its follow-up training. We see relatively good performance of Agreement variant in Door Key across all noise levels.

**2. Ablation based on labelling agent transitions using the classifier**. The variants considered within this study are (1) **Pseudo Label Argmax**: use the pretrained classifier to predict the feedback for unlabelled transitions and select the most probable feedback label to augment the replay buffer; (2) **Pseudo Label Sample**: use the pretrained classifier to predict unlabelled transitions and sample labels according to predicted probability distribution to augment the replay buffer. (3) **Pseudo Label Sample (Online)**: use the classifier to predict unlabelled transitions and sample labels according to probability distribution along with online training and (4) **Pseudo Label Argmax (Online)**: use the classifier to predict unlabelled transitions and sample the most probable labels along with online training of the classifier

The results for this ablation are shown in Figures 22, 23 and 24 respectively. As seen from the figures, Pseudo Label Argmax and Pseudo Label Sample shows significantly worse performance in comparison to CANDERE-COACH; showing no learning for higher noise (40%) suggesting that the pretrained classifier cannot predict the ground truth feedback well. The same trend is noticed for both the online variants as well which does not learn with 40% noise, highlighting that the labelling by the classifier induces more noise in the data. However, the classifier can still contribute to denoising based on the small loss trick as can be seen from the performance of CANDERE-COACH which outperforms the ablation variants significantly for all domains across all noise levels.

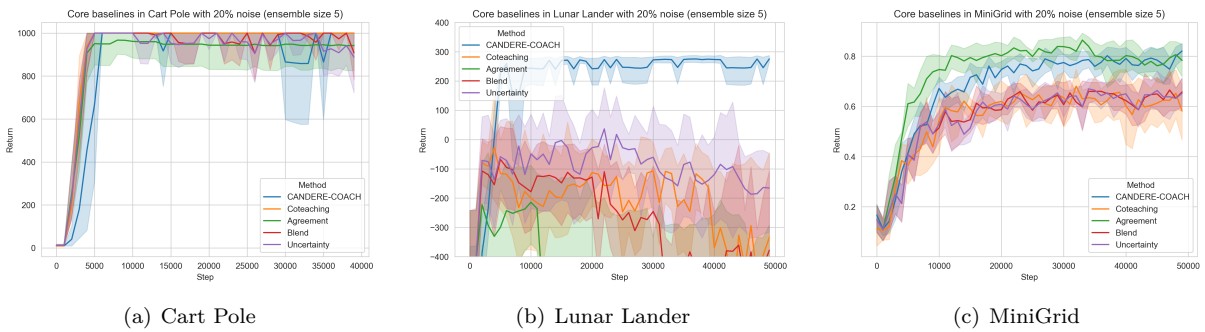

(a) Cart Pole         (b) Lunar Lander         (c) MiniGrid

Figure 19: Ablation results of uncertainty-based variants under 20% feedback noise in Cart Pole, Lunar Lander, and MiniGrid Door Key.

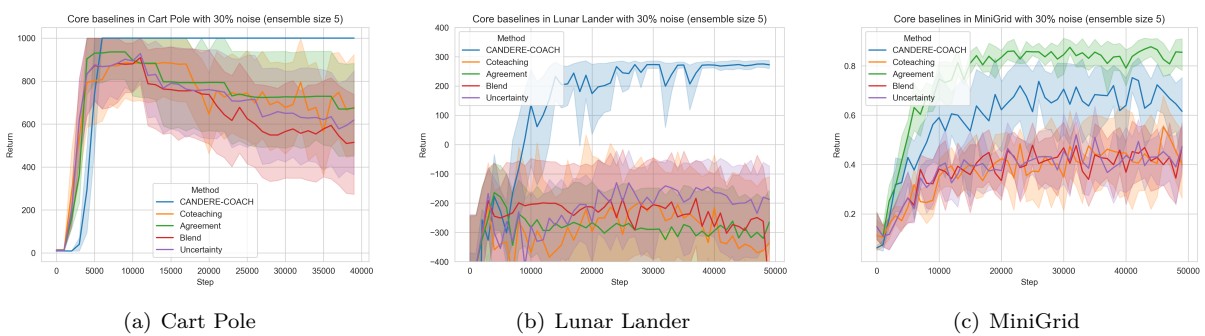

(a) Cart Pole         (b) Lunar Lander         (c) MiniGrid

Figure 20: Ablation results of uncertainty-based variants under 30% feedback noise in Cart Pole, Lunar Lander, and MiniGrid Door Key.

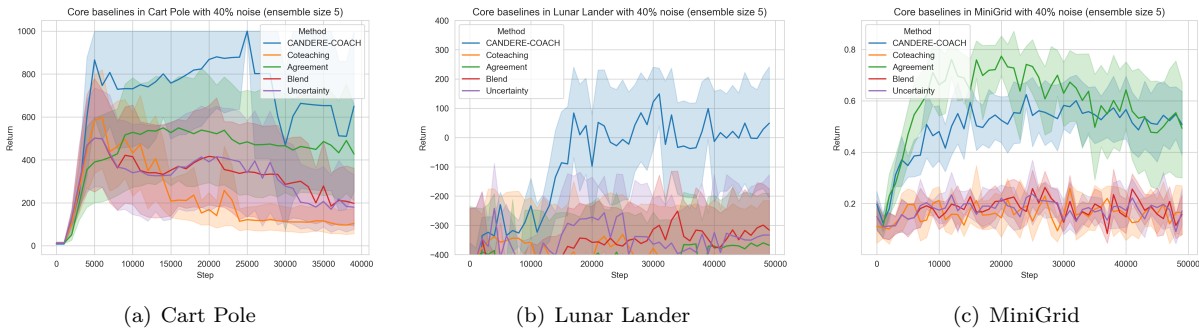

(a) Cart Pole         (b) Lunar Lander         (c) MiniGrid

Figure 21: Ablation results of uncertainty-based variants under 40% feedback noise in Cart Pole, Lunar Lander, and MiniGrid Door Key.

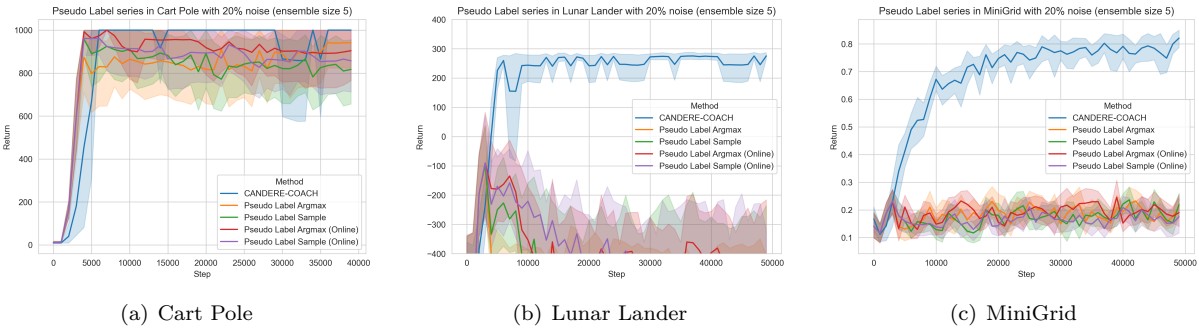

(a) Cart Pole      (b) Lunar Lander      (c) MiniGrid

Figure 22: Ablation results of pseudo label variants under 20% feedback noise in Cart Pole, Lunar Lander, and MiniGrid Door Key.

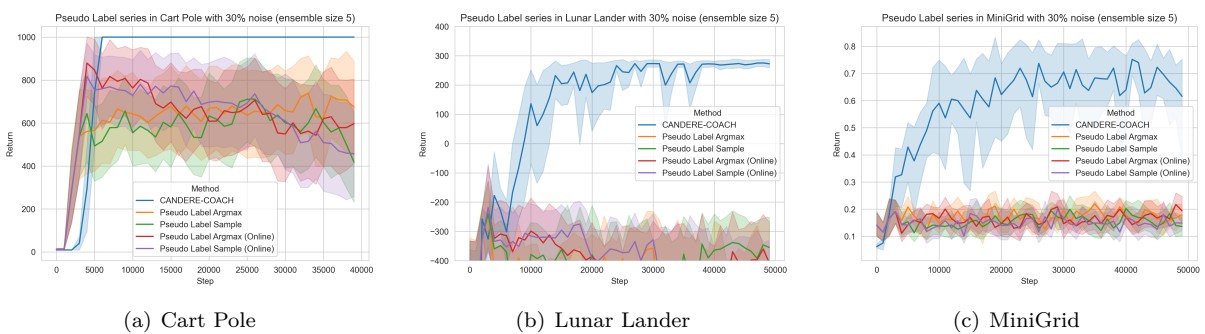

(a) Cart Pole      (b) Lunar Lander      (c) MiniGrid

Figure 23: Ablation results of pseudo label variants under 30% feedback noise in Cart Pole, Lunar Lander, and MiniGrid Door Key.

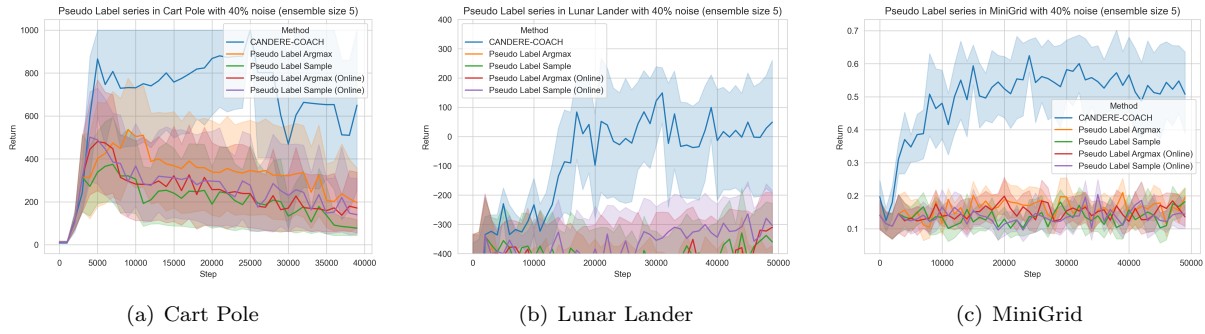

(a) Cart Pole      (b) Lunar Lander      (c) MiniGrid

Figure 24: Ablation results of pseudo label variants under 40% feedback noise in Cart Pole, Lunar Lander, and MiniGrid Door Key.

