# OpenReview forum: "CANDERE-COACH: Reinforcement Learning from Noisy Feedback"
_TMLR — Decision pending for TMLR_

### Review · Reviewer_sXcU · 2026-04-15

**Summary Of Contributions:**

This paper introduces an algorithm for RL from noisy feedback. The algorithm works by (1) training a detector for noisy feedback, (2) using that detector to identify noisy samples of feedback, and (3) relabeling samples deemed most likely noisy, and (4) training the model on feedback using COACH.

Overall, my review is negative, and i will not recommend acceptance. But I also want to say what I like about this paper -- I think that this paper communicates very well. I do not think that its contributions are novel w.r.t. past work, but I do think that the paper was conceived well and cleverly. Depending on the situation, I might recommend that the authors move on to new work instead of extending on this one, but I think that the paper has been a win from the perspective of insights being applied and tested well.

**Additional Comments:**

None

**Audience:**

No

**Audience Explanation:**

I think that the assumptions made in 4.1 prevent the paper from being useful. It is enormously convenient and unrealistic to assume assumptions 1-3. For assumption 1, the noise will depend on how good human evaluators are at assessing performance on a particular task. For example, humans will be great at knowing when models give correct info about basic facts but not at knowing when models give correct info about complex math concepts. For assumption 2, the proportion of flipped labels will generally not be known (though I see that this is relaxed later). Finally for assumption 2, the ground truth feedback will often not really exist since humans often disagree about important things in what models should do.

Overall, assumption 1 I think is the biggest issue. It trivializes the problem since scaling alone could solve any issue in which the noise distribution is homoskedastic and symmetric.

The paper does not use complex RL environments. The experiments here seem more typical of a paper from 2018 than one from 2026. So I do not think that is it is likely to be very impactful.

**Claims And Evidence:**

No

**Claims Explanation:**

I'm confused by the algorithm. Where are the rewards coming from? Where is the reward model? Isn't the classifier used just a reinvention of the reward model used in vanilla RLHF?

On the point of active relabeling, I don't see this paper doing much to engage with the active learning literature. A related active learning technique (one that doesn't have the downside of accidentally mislabeling when the reward model generalizes incorrectly) has been used in RLHF setups for around a decade.

I think it is somewhat unusual and confusing to say this is a apper about reinforcement learning. I see how the COACH algorithm is used, but I don't see a clear reason for why COACH should be. And if we are using coach to train a model simply using human-generated rewards and an RL-inspired loss, it's more a form of supervised learning than RL. I don't really understand why the COACH approach here is better than the many other algorithms that could be used.

The paper does not use baselines other than COACH.

**Requested Changes:**

The abstract does a good job of what the paper accomplishes and why, but says little as to how.

---

> ### Author Response · Authors · 2026-05-05
>
> We thank the reviewer for their constructive feedback. We respectfully ask the reviewer to reconsider the score if the clarifications below resolve their concerns.
>
> # Clarification on Algorithm Design
> For COACH [Macglashan et al., 2017 ] and it’s deep variant Deep COACH [Arumugam et al., 2019], the core assumption is that feedback is used as a surrogate for the advantage function; hence it can be used directly to guide the policy gradients. In these algorithms, there isn’t an explicit reward model; the human (like) feedback is used directly as a proxy for the advantage function as in Equation 1 of our paper.
> The reviewer's intuition is partially correct — our classifier does perform a function analogous to a reward model in vanilla RLHF; it estimates the feedback function from annotated feedback data.  However, it serves an additional role that distinguishes it from vanilla RLHF: it simultaneously identifies potentially corrupted feedback samples and corrects them via active relabelling, rather than treating all feedback as ground truth. This noise-awareness is not a feature of standard RLHF reward models.
>  This line of literature has similarities with RLHF with the following difference: (1) Human-in-the-loop RL methods like COACH and TAMER existed in RL literature even before RLHF was formalized in the context to LLMs; (2) reward function in our problem is learnt from scratch from the noisy feedback dataset rather than fine-tuned from a pretrained language model; (3) our setting targets sequential decision-making domains, where the feedback is over state-action pairs, not prompt-response pairs as in typical RLHF pipelines.
>
>
> # Active Relabelling Discussion
>
> As mentioned above, the scope of our work is not the RLHF line of literature that uses LLMs to do supervised training and then fine-tunes a reward model from preferences. We were inspired by uncertainty-based active learning techniques and have cited works from active learning literature like QACTOR [Younesian et al. 2021], where relabelling techniques have been used. We have added a paragraph in Related work to make this difference clear. We are happy to incorporate additional references if the reviewer can let us know the specific technique they are referring to.
>
>
> # Reinforcement Learning Nature of our Method
>
> The characterization of our method and COACH as supervised learning is not accurate. COACH as well as our method (since it is based on COACH) operates in an interactive, online loop where the agent takes action, receives feedback on those actions by a teacher/human and updates its policy using the received feedback. This is a typical RL setting where the data distribution is non-stationary because of the evolving policy thus making the feedback distribution non-stationary as well.
>
> We considered COACH to build our work on because our primary contribution is to study the impact of noisy feedback in established human-in-the-loop RL algorithms where this dimension of noisy feedback has been less studied. We also show that the proposed framework can be extended to other well-known human-in-the-loop RL algorithms like TAMER [Warnell et al. 2018].
>
>
> # Baselines other than Deep COACH
>
> Deep COACH serves as our primary baseline because our contribution is not a new RL algorithm — it is a noise mitigation framework that sits on top of existing human-in-the-loop RL algorithms. The natural baseline is therefore the same algorithm without noise mitigation, which is exactly what Deep COACH provides. To further demonstrate that our framework is algorithm-agnostic, we additionally show it generalises to TAMER, another established human-in-the-loop algorithm, and observe consistent improvements. We have added a line in the Baselines (section 5) paragraph to make this clear. We further introduced new baselines in Appendix Section I of Co-Teaching[Han et al 2018], Uncertainty, Agreement and Blend, where we show the CANDERE-COACH still generally shows better performance.
>
>
>
> # Assumptions in noise settings
>
> We acknowledge the assumptions as pointed out by the reviewer and mentioned in our paper. (1) Regarding assumption 1; we agree that in real-world scenarios, complex noise models would come into play based on expertise; however this is a reasonable assumption and have been made by many existing work along Learning from Preference [Cheng et al. 2024; Huang et al. 2025] and RLHF [Ouyang et al., 2022]. Extending a variant of this work for heteroskedastic noise models is our immediate future work.
> (2) As correctly pointed out by the reviewer, we relax assumption 2 later on. We do not assume access to ground truth feedback except for evaluation.

---

> > ### Author Response · Authors · 2026-05-05
> >
> > # Complexity of Environments
> >
> > We acknowledge this limitation in the conclusion. Our primary contribution is to study the impact of noisy feedback on established human-in-the-loop RL algorithms and to propose a framework to mitigate it. This is orthogonal to environment complexity, and discrete-action domains are sufficient to isolate and evaluate this effect. Extending to more complex environments is a natural direction for future work.
> >
> >
> > # Reference:
> > Cheng, Jie, et al. "Rime: Robust preference-based reinforcement learning with noisy preferences." arXiv preprint arXiv:2402.17257 (2024).
> > Wang, Yufei, et al. "Rl-vlm-f: Reinforcement learning from vision language foundation model feedback." arXiv preprint arXiv:2402.03681 (2024).
> > Ouyang, Long, et al. "Training language models to follow instructions with human feedback." Advances in neural information processing systems 35 (2022): 27730-27744.
> > Han, Bo, et al. "Co-teaching: Robust training of deep neural networks with extremely noisy labels." Advances in neural information processing systems 31 (2018).

---

> ### Author Response · Authors · 2026-06-10
>
> Dear Reviewer,
> Thank you again for your time and thoughtful comments. We have posted our rebuttal addressing the points you raised. We'd be grateful if you could let us know whether it adequately addresses your concerns, and whether any additional information or clarification would be helpful. We're happy to provide more details or run further experiments if useful.
> Thank you for your consideration.
>
> Best regards,
> The Authors

---

> > ### Comment · Reviewer_sXcU · 2026-06-10
> > **I will unfortunately weight in to reject**
> >
> > Hi, sorry for inaction for a while. It's a much busier time than normal. Thanks for the reply, but I do not think that the paper seems ready to be impactful in its current form. Independent of anything else, I am fairly persuaded that the troubles with assumptions from 4.1 alone.

---

> > > ### Author Response · Authors · 2026-06-11
> > >
> > > Thank you for your response.
> > >
> > > We would want to reiterate that while homoskedastic noise (standard random noise model) is a relatively simple model, it is a reasonable assumption and widely used in preference-based RL literature [Cheng et al. 2024; Huang et al. 2025], robust RLHF literature [Chowdhury et al. 2024; Wang al. 2024; Wu et al. 2025 ], as well as traditional ML literature [Han et al. 2018; Jiang et al. 2018;  Li et al. 2020] to study algorithm robustness against noise. Since our primary contribution is to propose a denoising algorithm inside learning from feedback in RL (COACH and TAMER-like algorithms), which has not been studied before, this work focuses on investigating the impact of homoskedastic noise models, with heteroskedastic noise models being our future work.
> > >
> > > # References:
> > >
> > > Cheng, J., Xiong, G., Dai, X., Miao, Q., Lv, Y. and Wang, F.Y., 2024. Rime: Robust preference-based reinforcement learning with noisy preferences. International Conference on Machine Learning (ICML), 2024
> > >
> > > Huang, S., Levy, M., Gupta, A., Ekpo, D., Zheng, R. and Shrivastava, A., 2025, May. Trend: Tri-teaching for robust preference-based reinforcement learning with demonstrations. In 2025 IEEE International Conference on Robotics and Automation (ICRA) (pp. 9574-9581). IEEE.
> > >
> > > Chowdhury, S.R., Kini, A. and Natarajan, N., 2024. Provably robust dpo: Aligning language models with noisy feedback. arXiv preprint arXiv:2403.00409.
> > >
> > > Wang, B., Zheng, R., Chen, L., Liu, Y., Dou, S., Huang, C., Shen, W., Jin, S., Zhou, E., Shi, C. and Gao, S., 2024. Secrets of rlhf in large language models part ii: Reward modeling. arXiv preprint arXiv:2401.06080.
> > >
> > > Wu, J., Xie, Y., Yang, Z., Wu, J., Chen, J., Gao, J., Ding, B., Wang, X. and He, X., 2025, May. Towards robust alignment of language models: Distributionally robustifying direct preference optimization. In International Conference on Learning Representations (Vol. 2025, pp. 16123-16150).
> > >
> > > J. Li, R. Socher, and S. C. Hoi, “Dividemix: Learning with noisy labels as semi-supervised learning,” In International Conference on Learning Representations (ICLR), 2020.
> > >
> > > Jiang, L., Zhou, Z., Leung, T., Li, L.J. and Fei-Fei, L., 2018, July. Mentornet: Learning data-driven curriculum for very deep neural networks on corrupted labels. In International conference on Machine Learning (pp. 2304-2313). PMLR.

---

### Review · Reviewer_paKq · 2026-04-17

**Summary Of Contributions:**

This paper studies the noisy label problem in feedback-based reinforcement learning. Drawing inspiration from anomaly detection, the authors propose pretraining a classifier on a noise-free dataset from an expert. During RL training, this pretrained classifier is used to provide rewards to the RL agent. By assessing the classifier's uncertainty (cross-entropy loss) regarding the label, the algorithm treats low-uncertainty labels as correct and flips those with the highest uncertainty. The authors term the proposed algorithm Classifier Augmented Noise DEtecting and RElabelling (CANDERE) and apply it to the COACH algorithm. Across three simple, standard Gym environments, the results demonstrate that CANDERE is beneficial, especially in high-noise settings and with a limited expert label budget.

### Strength
- The paper is mostly well-written, with the problem clearly motivated and the method well-explained.
- The experiments are well-designed to showcase the issue of noisy labels and how the proposed algorithm can address it.
### Weakness
- As acknowledged by the authors, the proposed algorithm has a few critical limitations (e.g., it only works for discrete action spaces, requires knowledge of the noise level, etc.).
- Although the experiments showcase the effectiveness of CANDERE, the selected environments are overly simple. There are potential issues that may arise in more complex situations, which are not explored in this paper.

**Audience:**

Yes

**Audience Explanation:**

Learning from expert feedback to replace standard rewards is an interesting direction in RL. I am sure there will be a broad audience within the RL community for this paper.

**Claims And Evidence:**

Yes

**Claims Explanation:**

The main claim of this paper is that by using a classifier to detect and correct noisy feedback, the RL algorithm can become more robust. The experimental results clearly support this claim.

**Requested Changes:**

There are a few details in the paper that I believe need further clarification or elaboration:

- In Section 4.2.1, the authors state there are two losses used in the paper, i.e., Eq. 2 and Eq. 3. However, I am confused about where each one is used. For example, there is no mention of Eq. 3 in Algorithm 1. Is the focal loss only used during pretraining?
- In Eq. 5, I believe the condition should be $|B^\prime| \leq R^\prime(B)|B|$; otherwise, we will have $B^\prime = B$ since having more samples always increases the total loss.
- The notation can also be a bit confusing; for example, the remember rate $R(B)$ and the relabelling rate $R^\prime(B)$. Better-designed notation would ease readability.
- The paper is unclear regarding the limited budget setup. In the main experiments, it mostly uses a limited budget of 1000 feedbacks within 250k environment steps in Figure 3(b) or 40k–50k steps in Figure 4. However, the hyperparameter section mentions a feedback frequency of 10. These numbers do not match. Additionally, Algorithm 1 does not seem to consider the limited budget setup, since you do not always have an $f$ from the replay buffer.

Some design choices regarding the relabelling are unclear to me, and the authors should elaborate on them:

- I noticed the relabelling ratio is very aggressive (i.e., 0.6 in the hyperparameter settings). Given that the remember rate $R(B) = 1 - p_{noise}$ is a minimum of 0.6, there will be some overlap between the sets $B_n$ and $B_c$. Is this by design?
- Since the classifier mainly uses its own output to train itself during the online phase, won't that lead to some form of degeneration or collapse? If not, what is the intuition preventing this?
- I believe the relabelling could potentially lead to problems (although this doesn't seem to present itself in the experiments). For example, it could happen that a random mini-batch only contains correct labels (all with low CE loss), but the relabelling mechanism still forces some labels to be flipped simply because their loss is slightly higher than the others. Is there a mechanism in place to prevent this, such as checking if the CE loss is lower with the flipped label? This would be worth discussing in the paper.

---

> ### Author Response · Authors · 2026-05-05
>
> We thank the reviewer for their constructive feedback. We respectfully ask the reviewer to reconsider the score if the clarifications below resolve their concerns.
>
>
> # Equations and Notations
> We thank the reviewer for pointing out the possibly confusing equation references. In our design, we may use both loss functions, in our practice, for pretraining, we used focal loss for potential class imbalance issues and for online updates we used standard cross entropy loss.
> We also thank the reviewer for pointing out the potential readability in our notations, we will improve this in our final version of the paper if accepted.
>
>
> # Clarification on Budget Setting
> Our algorithm is designed to operate in both limited- and unlimited-budget settings. In the unlimited-budget setting, however, its relative advantage is expected to be less significant: when the feedback noise rate is below 50%, sufficiently many interaction steps allow the learner to mitigate noise through repeated feedback collection. In our experiments, feedback is queried every 10 environment steps under a budget of 1,000 queries, which is exhausted after 10,000 steps. After the budget is depleted, the agent continues learning only from the feedback accumulated in the replay buffer.
>
>
> # Relabelling ratio and potential problems
> As pointed out by the reviewer, an extremely high relabelling ratio can be problematic. However, B_n and B_c are never overlapping with each other. The relabelling is only conducted on B_n (noisy set) and relabelling ratio means the percentage of labels flipped inside B_n.

---

> > ### Comment · Reviewer_paKq · 2026-05-20
> >
> > Thanks the authors for providing rebuttal for my questions.
> >
> > Although the rebuttal clears out a few details I think there are still a few confusing parts regarding the training procedure. I am looking at the algorithm 1.
> > - In line 6, $B= \{(s_t, a_t, f_t), ... \}$ are sampled from the replay buffer. But in the limited budget setup, not all the data point are labelled by the expert so some of the data in the replay buffer won't have labels. Do you only sample those data with the labels or you sample from all the data but just some of them without labels? I would assume the later case, otherwise it won't make sense to make further interactions with the environment after the budget is used up.
> > - In line 7, a new label is computed by the classifier. Do you use this label to overwrite with the label from the buffer or do you only compute label for those don't have labels? In the first case, when will the real label get used?
> >
> > I believe these details are very important to understand what the later parts of the algorithm operating on.

---

> > > ### Author Response · Authors · 2026-05-20
> > >
> > > We thank the reviewer for their comments.
> > >
> > > Line 6: As the reviewer pointed out, not all data points are labeled by the expert or noisy teacher. In our method, we only sample data points for which a label was provided.
> > >
> > > Line 7: We do not use the newly predicted label to overwrite the original label. Instead, we use the classifier’s predicted logits to compute the cross-entropy loss, which enables the small-loss trick for detecting noisy labels. We also conducted an additional ablation study in Appendix I with a new baseline called Agreement. In this baseline, we filter out labels for which the classifier’s prediction does not match the received label, and only learn from examples where the classifier agrees with the teacher’s label. The ablation results suggest that although a pretrained classifier may not always predict labels correctly, it can still help distinguish clean from noisy labels through the small-loss trick.

---

> > > > ### Comment · Reviewer_paKq · 2026-05-20
> > > >
> > > > Thanks for your clarification.
> > > >
> > > > This is a surprising answer. If it is correct, that means with a budget size of 1000, the agent is only trained on 1000 data points, and most of the time the agent is doing offline RL after the budget is used up in the first 10k steps (as no more data will be fed into the buffer). It is surprising that a simple objective in Eq. 1 can solve the tasks on image observations with only 1000 samples. In this case, I am curious:
> > > > - Why does the agent still interact with the environment after the budget is used up? It doesn't seem to help with the training in the current setup.
> > > > - Will the algorithm perform even better if we also use the classifier to provide labels for those unlabelled data?
> > > >
> > > > By the way, there should be a small correction following your previous response regarding the relabelling. If, as you mentioned, there is never an overlap between $B_c$ and $B_n$, but in Table 6 the active relabelling ratio $R^\prime (B)$ is set to 0.8 when the noise level is 0.4, there will be a 20% overlap between $B_c$ and $B_n$. So maybe there is a typo, or the relabelling actually happens on $B \setminus B_c$ in Eq. 5?

---

> > > > > ### Author Response · Authors · 2026-05-21
> > > > >
> > > > > We thank the reviewer for their prompt response.
> > > > >
> > > > >
> > > > > # Observation Space Clarification
> > > > > The observation space is not purely image-based in MiniGrid Door Key. We used a pretrained CNN network to extract features to downsize the dimension. The feature extractor details are presented in Appendix F.
> > > > >
> > > > > # Budget depletion clarification
> > > > >
> > > > > As correctly pointed out by the reviewer, we do offline RL with the collected feedback data once the budget is depleted, without collecting more agent experience online. However, after budget depletion, the joint training of the classifier and the policy continues. The subsequent training of the classifier with data from the replay buffer (all annotated) helps it to train further and identify potential noisy points. Therefore, it is meaningful to continue the training steps to help the classifier refine the choice of potential noisy feedback labels and hence, make the RL policy update noise-robust. Another point to note is that the feedback labels corresponding to the collected states and actions might change even after budget depletion, depending on whether they are used in the active-relabelling step or not.
> > > > > In particular, the classifier continues to be trained using the fully annotated data stored in the replay buffer. This continued training enables the classifier to further refine its predictions and identify potentially noisy labels. As a result, extending the training process is necessary beyond budget depletion, as it improves the classifier’s ability to detect noisy feedback and consequently enhances the robustness of the RL policy update.
> > > > > Additionally, it is important to note that feedback labels associated with previously collected state-action pairs may still change after budget depletion. This occurs when such samples are selected during the active relabelling phase, allowing further correction and refinement of the replay buffer data.
> > > > >
> > > > > Here is the response to the specific questions in this regard:
> > > > > - Algorithm 1 is for the unlimited budget setup, hence the confusion. We have updated the algorithm with the limited budget setup in the current version.
> > > > > - While using a classifier to predict feedback seems feasible, this requires a well-trained model (i.e. abundant pretraining data) so that the overall noisy feedback proportion does not increase steeply in the process. If we have enough pre-training data to train a classifier to accurately predict whether an action is good or bad, it can directly act as a policy for the task (by iterating over the actions and executing one that has the highest positive feedback distribution). Our problem setting is specific, where we don’t have enough data to train a good classifier (e.g., a budget of 1000 and pretraining data size of 25/30 in cart pole); however, their loss pattern can still help denoising without the classifier being particularly good in the supervised learning task itself.
> > > > >
> > > > >
> > > > >
> > > > > # Overlapping of different sets of labels
> > > > > The reviewer is correct that relabelling happened in $B \setminus B_c$. We added a footnote in the paper (Page 5) to clarify this.
> > > > > Taking the reviewer’s example, if we have a relabelling ratio of 0.8 and noise of 0.4, then we detect 40% of noisy labels, and within these 40%, potential noisy labels, we flip 80% of them. In other words, in total, we have 32% (0.8*0.4) of labels flipped. Equation 5 is meant for $B_n$ (the detected noisy batch), and the active relabelling batch $B_{ar}$ is defined inside $B_n$. This is mentioned in Section 4.2.3.

---

> > > > > > ### Comment · Reviewer_paKq · 2026-05-21
> > > > > >
> > > > > > Thanks for answering the further questions.
> > > > > >
> > > > > > First, regarding the relabel, yet it is another discrepency from what is current written in the paper. Following your explaination, we should have $B_n = B \setminus B_c$ and $R^\prime(B)$ is not a hyper-parameter in Eq. 5. As we have already found a lot of discrepancies between what is written in the original draft and what is explain and confirmed by you during this rebuttal, I would urge the authors to re-evaluate what they write in the paper and what they actually do in the implementation before any further discussion. And a code needs to be submitted for the final version.
> > > > > >
> > > > > > Regarding the offline training, I understand what you did but the current paper has no mention about it, which makes it very confusing. Maybe consider to explain it in the paper and draw a vertical line in the plots to indicate when does the offline training start. But even in this case, having x-axis as the environment steps but don't use it is still counterintuitive.
> > > > > >
> > > > > > Regarding the additional experiments about labeling the unlabelled data with the classifier, I still think it is benefit to do it. Your argument is only one of the possibilities where you assume the classifier is not good enough. But as you have already chose very simple problems, it is also possible that the classifer trained on this 1000+ data points is perfect enough already. Depends on the results, the conclusion will be different about whether the proposed algorithms can handle non-perfect classifer.

---

> > > > > > > ### Author Response · Authors · 2026-05-26
> > > > > > >
> > > > > > > We thank the reviewer for their helpful suggestions and feedback. We are currently revising the paper to improve content as suggested, and we are also conducting the additional experiments requested in the review. We appreciate the reviewer’s patience and will post a more detailed updated response once these revisions and experiments are complete by the end of the week. We will also submit the code in the final version as suggested by the reviewer.

---

> > > > > > > > ### Author Response · Authors · 2026-06-01
> > > > > > > >
> > > > > > > > We thank the reviewer for their patience and for pointing out these issues. We have carefully updated the paper with all **changes highlighted in blue** to clarify specific technical details as pointed out by the reviewer. Below are the specific changes
> > > > > > > >
> > > > > > > > **On clarification about details on active relabelling**: We would like to clarify that the previous notation in Eqn 5 was a general one where the R’(B) can be a hyper-parameter as in prior work [Han et al. 2018]. We intended to allow a more general formulation in which (R(B)) and (R'(B)) do not necessarily sum to one, since in practice one may apply either a more conservative or more aggressive denoising window. For our implementation, as pointed out by the reviewer, we use a value of R’(B)  equal to the noise rate.
> > > > > > > >
> > > > > > > > In the revised version (section 4.2.2 & 4.2.3 highlighted in blue), we have updated the description of the relabeling procedure and clarified the details of the clean and the noisy sets in Eq. 5. We have revised the notation and surrounding explanation to make this point explicit and to improve readability.
> > > > > > > >
> > > > > > > > **Final Code submission**: We will submit the code with the final version and make it open-source.
> > > > > > > >
> > > > > > > > **Offline Training details**: Regarding offline training, we have revised the relevant sections (Section 4 Algorithms, Section 5 Baselines, Section 5.1 CANDERE-COACH evaluation) to describe this procedure clearly. We have updated the figures by adding vertical markers indicating the point at which the feedback budget is depleted and offline training starts. We also changed steps to iterations in our plots to accurately reflect the learning settings. We also clarify this in the caption and main text.
> > > > > > > >
> > > > > > > > **Augmenting the unlabelled transitions with the classifier predictions**: Following the reviewer’s suggestion, we have also conducted additional experiments that use a pretrained classifier to label the unlabeled data. Specifically, we introduce two new variants: **Pseudo Label Argmax** and **Pseudo Label Sample**, with results reported in Appendix I (also contains other ablations suggested by reviewer ucdX). Pseudo Label Argmax assigns each unlabeled example the classifier’s most likely feedback label, while Pseudo Label Sample samples from the classifier’s predicted label distribution for augmentation. These experiments help test whether the classifier can directly augment the unlabeled dataset, rather than only being useful for denoising. The results show that, in our setting, directly pseudo-labeling unlabeled data does not improve performance, whereas the classifier remains useful for identifying noisy feedback. This suggests that the proposed method can benefit from imperfect classifiers, even when their predictions are not reliable enough to serve as direct labels.
> > > > > > > >
> > > > > > > >
> > > > > > > >
> > > > > > > >
> > > > > > > >
> > > > > > > > **References**:
> > > > > > > > Han, Bo, et al. "Co-teaching: Robust training of deep neural networks with extremely noisy labels." Advances in neural information processing systems 31 (2018).

---

> > > > > > > > > ### Comment · Reviewer_paKq · 2026-06-08
> > > > > > > > >
> > > > > > > > > Thanks for making the corrections to the draft; the current version is much clearer.
> > > > > > > > >
> > > > > > > > > The new experiments re-emphasize the findings in Figures 6 and 7, demonstrating that online training is crucial to the final performance. Although an ideal experiment would use the online training classifier to label the replay buffer on the fly, I think that would be something extra and independent of the correctness of the current claim.

---

> > > > > > > > > > ### Author Response · Authors · 2026-06-12
> > > > > > > > > >
> > > > > > > > > > We thank the reviewer for recognising our improved draft and newly added experiments. We further added Pseudo Label Sample (Online) and Pseudo Label Argmax (Online) as suggested by the reviewer in Appendix I (Figures 22, 23 and 24) of the updated draft. As suggested by our results, the performance doesn’t change much in comparison to Pseudo Label Sample and Pseudo Label Argmax, and is still outperformed significantly by CANDERE-COACH. This suggests that with online training, the imperfect classifier’s ability to predict feedback does not improve over time, as many incorrect labels are injected, resulting in more added noise.
> > > > > > > > > >
> > > > > > > > > >  For convenience, we have now split the ablations based on 2 criteria: (1) uncertainty-based estimates suggested by Reviewer ucdX (Fig 19, 20, 21) and (2) classifier labelling-based ablations as suggested by you (Fig 22, 23 and 24).

---

### Review · Reviewer_ucdX · 2026-04-20

**Summary Of Contributions:**

The paper investigates a reinforcement learning setting with noisy feedback. The authors argue that traditional methods like DEEP-COACH operate under the assumption that teacher feedback is perfect, and hence do not perform as well in practice under noise. To alleviate this, the authors propose a filtering mechanism that denoises data in an online manner, and updates the classifier online by using the most-trusted data or the least-trusted ones, flipped. Experiments on three Gymnasium benchmarks (Cart Pole, Lunar Lander, and Minigrid Door Key) show improved performance compared to standard methods like DEEP COACH or DEEP TAMER.

**Audience:**

Yes

**Audience Explanation:**

The topic of reinforcement learning with noisy feedback that this paper's dealing with is interesting to a wider TMLR audience, especially given the recent rise of RLHF, DPO from human data, and so on.

That said, as I explain in the last sentence of the previous section, the fact that the authors essentially do an extension of rather old works from 2018 and 2019 without a more modern experimental evaluation (e.g., with RLHF) may make this work less relevant and interesting that it could have otherwise been.

**Broader Impact Concerns:**

No broader impact.

**Claims And Evidence:**

No

**Claims Explanation:**

Strengths
The paper topic is interesting and well-motivated. The paper is mostly clear. The authors have performed ablations to understand (i) the effect of online training on noise-filtering classifier, as well as (ii) the effect of noisy pretraining dataset. I also like the use of a focal loss. The algorithm is simple but the methodology looks sound.

All that said, I feel there are various weaknesses:
- The algorithm is simple and the authors did not investigate more options. For instance, the authors could have used ideas from epistemic uncertainty, e.g., with an ensemble of different classifiers and the quantification of uncertainty over the different samples. I think the current framework is interesting, but in the context of noisy feedback, I was thinking the authors could have developed more methods. It is possible their current algorithm would have still performed the best, but it'd still be interesting to see how other methods would work.
- In this framework, labels are flipped with a given fixed probability $p_{noise}$. This sounds quite restrictive. In practice, some state-action pairs $(s, a)$ may have much more noise. Also, noise may be different based on how the original teacher was pretrained. Maybe it will be noisier in certain subspaces but less noisy in others. The authors did not discuss that point at all. I feel the paper would have benefitted from investigating more modelling assumptions and/or algorithms regarding the noise, even extensions of the current framework.
- Deep COACH has considered 3 values for the feedback , i.e., {-1, 0, +1}. The current work only considers possible values {-1, +1}. I was wondering what motivated this choice and whether it restricts the framework in any meaningful way.
- The 3 benchmarks are interesting. But I was feeling that this framework would be particularly useful in the context of RLHF (e..g., DPO), where human feedback is provided, often in a noisy form. The authors do mention human feedback in the Introduction, but experimentally they do not focus on it at all. It would have been interesting for example to show that practical RLHF with noisy feedback (e.g., due to the noisy data collection) can benefit from the proposed framework.
- Even though the proposed methods perform on average better, their variance is extremely high, often even with a small amount of noise of 10%. Is that due to limited iterations? Would variance drop if the authors did many more iterations with multiple initialisations? The current variance is very high, and this could possibly point to lack of stability or robustness of the proposed framework (e.g., sometimes it performs well, but some others poorly). Is it possible to reduce that variance, at leats for lower values of $p_{noise}$?
- Deep COACH and Deep TAMER are rather old papers (from 2018 and 2019, if I recall correctly). Essentially, this paper is proposing an extension of these methods, if I understand correctly. Why focus on such old works and not more recent ones? I understand the argument that these works may be very relevant today in the context of RLHF, but then the authors never actually experimented with RLHF and related methods. This then creates the question: is this work truly relevant and interesting?

**Requested Changes:**

I'm open to revising my score upward, but I would ask the authors to address all weaknesses and concerns or questions I previously listed. In those cases where new experiments or theory are not a possibility, I'd be still interested to get the authors' feedback and counterarguments.

---

> ### Author Response · Authors · 2026-05-05
>
> We thank the reviewer for their constructive feedback. We respectfully ask the reviewer to reconsider the score if the clarifications below resolve their concerns.
>
> # Uncertainty Baselines
>
> We thank the reviewer for suggesting potential baselines. We introduced 4 more baselines:
> Co-Teaching: classifier start from scratch, following the previous work in [1] and use the small loss trick
> Uncertainty: we initialise an ensemble of pretrained classifiers (size=5) and use the variance to predict clean or noisy labels
> Blend: Similar to uncertainty but use a mixed score of ensemble variance and small loss trick
> Agreement: instead of using loss as an indicator, we only learn from samples where the classifier predict the same with the received label (i.e. the classifier agrees with the label)
> The figures are presented in the newly added Appendix I (changes highlighted in blue). CANDERE-COACH still generally out performs other methods. More information are reported in the updated paper draft, Appendix I.
>
> [1]Han, Bo, et al. "Co-teaching: Robust training of deep neural networks with extremely noisy labels." Advances in neural information processing systems 31 (2018).
>
>
> # Fixed probability
> While fixed probability doesn’t sound necessarily realistic, many previous work such as BPref, RIME, Co-teaching [1,2,3] has used this to induce noise. Furthermore, though feature-dependent noise can be a realistic model to heteroskedastic noise (e.g. noise percentage depending on state features), previous work showed that general denoising algorithms might not be robust to all types of FDNs [4]. Therefore, our primary focus stays on one type of noise: uniform noise (e.g. accidental flipping) to study the proposed noise-aware framework.
>
> References:
>
> [1]Lee, Kimin, et al. "B-pref: Benchmarking preference-based reinforcement learning." arXiv preprint arXiv:2111.03026 (2021).
>
> [2]Cheng, Jie, et al. "Rime: Robust preference-based reinforcement learning with noisy preferences." arXiv preprint arXiv:2402.17257 (2024).
>
> [3]Han, Bo, et al. "Co-teaching: Robust training of deep neural networks with extremely noisy labels." Advances in neural information processing systems 31 (2018).
>
> [4]Li, Yuxuan, Harshith Reddy Kethireddy, and Srijita Das. "Evaluating Feature Dependent Noise in Preference-based Reinforcement Learning." arXiv preprint arXiv:2601.01904 (2026).
>
> # Disregarding Indifferent Feedback
>
> For COACH, a feedback of 0 leads to policy gradients; the update equation is as per Eqn -1. So, essentially, if f_t=0; gradient is 0. Therefore, we disregard the indifferent feedback.
>
> # High Variance
> We thank the reviewer for their scrutiny. In Figure 5, the experiment is done with noisy pretraining dataset (i.e. classifier trained from noisy data at the beginning), hence the variance is high
> In contrast, In Figure 17, We have relatively low variance (both A/R and no A/R).
>
>
>
> # Difference with RLHF
>
> Our problem can be extended to the RLHF setting which is our immediate future work. However, RLHF can be a different domain,  because we are training a control policy from scratch as compared to RLHF problems that fine-tune a reward model for alignment on a pre-trained model.
>
> # Why binary feedback
> Binary feedback is intuitive and easy to provide, hence the choice [1]. This is different from Preferences in the RLHF kinds setting where preference is over trajectory segments, which requires higher ability to distinguish.
>
> References:
> [1]Zhong, Victor, et al. "Policy improvement using language feedback models." Advances in Neural Information Processing Systems 37 (2024): 43730-43758.
>
> # Why COACH/TAMER?
> We want to study the impact of noisy feedback on traditional RLHF models in literature other than LLMs which has not been sufficiently explored. Recent RLHF based decision-making problems are often positioned in NLP and LLM applications, while our domain focuses on sequential decision-making domains. While LLMs, especially thinking models, possess planning ability, they are not computationally efficient and traditional RL algorithms still fit better.

---

> > ### Comment · Reviewer_ucdX · 2026-06-09
> > **additional baselines**
> >
> > I thank the authors for addressing my concerns. The new baselines seem to significantly underperform the proposed method (with the occasional exception of the Agreement baseline). This definitely looks very nice. One follow-up question though: Do these baselines assume a clean, noisy-free pretraining dataset to build the classifiers used, e.g., in Co-Teaching, and Uncertainty? Or were these baselines trained differently?
> >
> > The reason why I feel this is important is that comparison for the different methods should be as fair as possible (e.g., in terms of the original pretraining, the online training etc.). How did the authors ensure that? Importantly, is the very clear advantage of the proposed method due to the superior denoising scheme only?
> >
> > I'd appreciate it if the authors could elaborate more on that. The results in Appendix are very encouraging, but I wasn't clear how exactly the comparison was made. It would help to understand where the very clear advantage of the proposed scheme came from exactly.
> >
> > Regarding my other concerns, I was covered by the rebuttal.

---

> > > ### Author Response · Authors · 2026-06-11
> > >
> > > We thank the reviewer for the useful feedback & suggestions and for acknowledging that several prior concerns have been addressed in the rebuttal.
> > >
> > > Regarding the newly added baselines, we would like to clarify that all the ablation algorithms in Fig 19, 20, 21 in the appendix also assume a noise-free pre-training dataset as CANDERE-COACH, ensuring a fair comparison among all. In addition, all experiments use the same pretrained classifier trained on the same dataset. For the ensemble-based methods (Uncertainty, Blend), each classifier in the ensemble is initialized from the same pretrained weights, and the ensemble models subsequently diverge during online training due to different sampled mini-batches.
> > >
> > > We have added these details to the appendix to further clarify the experimental setup.
> > >
> > > We believe the advantage comes from using the denoising scheme. Additionally, the ablation variants suggested by reviewer paKq in the same figure (Pseudo Label Argmax and Pseduo Label Sample) show that although a pretrained classifier may not always predict labels correctly (the classifier was used to predict feedback labels of agent transitions), it can still help distinguish clean from noisy labels through the small-loss trick property of Neural Networks.

---

### Decision · Action_Editor_YBcu · 2026-06-22

**Recommendation:** Accept as is

**Audience:**

Yes

**Audience Explanation:**

The findings of this paper are relevant to researchers and practitioners in reinforcement learning with expert feedback and preference-based learning. The proposed methodology may also be applicable to related paradigms, such as RLHF and DPO, where robustness to noisy human feedback is an important consideration.

**Claims And Evidence:**

Yes

**Claims Explanation:**

The paper studies a reinforcement learning framework with noisy feedback. The authors note that existing approaches, assume that teacher feedback is error-free, which can significantly degrade performance in realistic noisy environments. To address this limitation, they introduce an online filtering mechanism that denoises feedback data and continuously updates a classifier using either the most reliable samples or the least reliable samples after label correction. Experimental results on three Gymnasium benchmarks, demonstrate consistent performance improvements over baseline methods.